# A Novel Electrophoretic Technique to Improve Metasurface Sensing of Low Concentration Particles in Solution

**DOI:** 10.3390/s23208359

**Published:** 2023-10-10

**Authors:** Zachary A. Kurland, Thomas Goyette

**Affiliations:** Submillimeter-Wave Technology Laboratory, University of Massachusetts Lowell, Lowell, MA 01854, USA; thomas_goyette@uml.edu

**Keywords:** metasurface, metasurface sensing, electrophoresis, nanoparticles, sensing, microwave sensors, materials science, millimeter wave devices

## Abstract

A novel electrophoretic technique to improve metasurface sensing capabilities of charged particles in solution is presented. The proposed technique may improve the ability of metasurfaces to sense charged particles in solution in a manner not possible using the current standard of particle deposition (which allows particles to sediment randomly on a metasurface under evaporation) by inducing an external, nonuniform electric field through the metasurface apertures. Such a technique may be useful in various sensing applications, such as in biological, polymer, or environmental sciences, where low concentration particles in solution are of interest. The electrophoretic technique was simulated and experimentally tested using latex nanoparticles in solution. The results suggest that, using this technique, one may theoretically increase the particle density within the metasurface regions of greatest sensitivity by nearly 1900% in comparison to random sedimentation due to evaporation. Such an increase in particle density within the regions of greatest sensitivity may facilitate more precise material property measurements and enhance identification and detection capabilities of metasurfaces to particles in solution which constitute only a few hundred parts per million by mass. It was experimentally determined that the electrophoretic technique enhanced metasurface sensing capabilities of 333 parts per million by mass latex nanoparticle solutions by nearly 1700%.

## 1. Introduction

Electromagnetic metasurfaces which are suitable for sensing applications typically consist of sub-wavelength structures which have been carefully engineered to enhance light-matter interactions in their vicinity. Recently, the exact characteristics which make a metasurface suitable for sensing applications have been offered in [1], which details how one may perturb a simple topology to produce an electromagnetic “hotspot” in some volume surrounding metasurface apertures. Such hotspots have been shown to enhance light-matter interactions between incident radiation and dielectric materials on metasurfaces, thus increasing their sensing capabilities [2,3].

Just as in [1], many of the sub-wavelength metasurface structures which have been engineered for sensing applications are small apertures etched in a thin metal film [4,5,6], and constitute a small fraction of the total surface area of a metasurface. Since the relative area of a metasurface’s apertures can be exceedingly small in comparison to the area upon which a material of interest in sensing efforts is deposited, it could be that much of a material is going undetected by the apertures; or at least, much of the material may settle or is deposited in regions where the material-aperture interaction is sub-optimal. In this paper, a novel electrophoretic technique is presented which seeks to address a shortcoming of metasurface topologies most suitable for the sensing of low concentration particles (as low as a few hundred parts per million by mass) in solution by attracting charged particles using electrophoresis. By inducing an external, nonuniform electric field through the metasurface apertures, charged particles in solution could be electrophoretically guided towards metasurface apertures, and sediment where their sensitivity to changes in their ambient environment is greatest.

One such field of metasurface sensing research which could benefit greatly from this electrophoretic technique involves using metasurfaces to sense water contaminants such as pesticides [6], bacteria [7], and viruses [4] in solution. Water contaminants, pose a major threat to people and environments all over the world. For example, an excess of nitrate ions in water may contribute to the formation of various cancers, Parkinson’s disease, and health problems in infants if ingested in excess [8]. Therefore, it is important that the levels of such toxins in water be monitored. There are multiple methods used to detect the many dangerous particulates in water, such as Potentiometric [9], Amperometric [10], Voltammetric [11], Planar Electromagnetic [12], Fiber Optic [13], and Metasurface [4,6,7] sensing techniques. However, current metasurface sensing techniques designed to detect dangerous charged particles in solution may not be living up to their full potential.

An electrophoretic metasurface sensing technique (which could force charged particles in solution to gather at metasurface apertures via an externally applied electric field), however, could increase the probability that dilute charged particles in solution are detected by a metasurface, and potentially lower the detection concentration threshold of a material of interest. This could be especially useful in the metasurface sensing efforts of pesticides [6,14], and some viruses which have been shown to move under the influence of electrophoresis [15].

## 2. Materials and Methods

The objective of the proposed electrophoretic technique is to force charged particles in solution to gather at the most sensitive metasurface sites such that, once the suspending solution evaporates, the settled, charged particles will be more easily detected [16,17]. The charged particle of interest in this study are colloidal latex particles. The reason why it is advantageous to study colloidal latices is detailed in Section 2.1. The metasurface fit for sensing applications (which will be the focus of this study) has been well described in [1]. The unit cell of this metasurface topology is provided in Figure 1, and its approximate dimensions given in Table 1.

In order to evaluate the efficacy of the electrophoretic technique, an experimental apparatus was devised, and is depicted in Figure 2. Using such an experimental setup, it was suspected that because of the potential difference produced between the metasurface and ground plate, a negative electric field strong enough to attract positively charged particles in suspension would emanate from the metasurface apertures. Since the efficacy of the electrophoretic technique was studied experimentally and through simulation, it is important to first establish an understanding of colloidal latices, the regions of greatest sensitivity on a metasurface, electrostatic fields through metasurface apertures, and the forces on charged particles suspended in an evaporating medium while subject to nonuniform electric fields.

### 2.1. Colloidal Latices

When discussing the different types of particles that exist within solutions that one may apply this technique to, particle size is an important parameter to discuss, as it dictates the types of forces that they may be subjected to. For instance, the number and scope of criterion which qualify a solution as a colloidal system is quite vast, ranging from fog and mist, to milk, to metallic hydroxides, silica gels, and bio-colloids such as blood, bone, muscle, and cells. Interestingly, there exists materials which exhibit all permutations of solid, liquid, and gas combinations that are considered colloidal systems, making them quite a broad set which can be explored. Furthermore, colloidal systems generally contain particles whose sizes range from 1 nm to 1 μm; though, the size of particles within the systems can vary dramatically, with some particles much larger than 1 μm [18,19].

One such colloid of interest is latex. Latex particles (latices) are prime candidates for study using this electrophoretic technique via W-band metasurface because of their relatively large particle size, surface charge, ability to sediment, and the amount of research that has been conducted on them. Unfortunately, latex formulations can contain a significant level of volatile organic compounds which can damage the ozone layer, whereas others are linked to greenhouse gasses which contribute to climate change. Latex, therefore, is an important material to study, since it is widely used in architectural coatings, textiles and carpets, various construction materials, paper coatings, inks, gloves (where 135 billion latex gloves were manufactured in 2008 alone), and many other over-the-counter items [20].

Rubber latex is generally referred to as a stable dispersion of hydrophobic rubber particles. In large part, it owes its stability to the charge carried by each of the particles within the colloidal suspension. Latex contains many charged species, such as ionic surfactants and salts [20]. The use of surfactants in latex is mainly due to their stabilising properties, and play a vital role in the formation of smooth latex films. In general surfactants are classified by their polar head groups, of which there are four different types. Particularly relevant to this study, cationic surfactants are comprised of positively charged groups such as long chain amines and their salts, quaternary ammonium salts, and polyoxyethylenated long chain amines [21].

When a colloidal dispersion (such as latices in water) is deposited onto a surface, water evaporation commences and forces the particles to sediment and pack into an often face-centered cubic lattice [22,23]. Vertical deposition processes, where the evaporation rate of the solvent is greater than the sedimentation velocity of the particles, is a highly effective method to deposit such colloids. During vertical deposition, convective mass flow and capillarity are the two main driving forces which enable colloidal particles to assemble on a substrate, producing films that can be several microns thick. Finally, it has been shown that patterns on the surface of substrates can be used to control the assembly of colloidal particles, and, most relevant to this study, that they will preferentially attach and assemble in regions of opposite charge [20].

### 2.2. Quantifying the Metasurface Regions of Greatest Sensitivity

In this subsection, a technique is presented in which the sensitivity of a metasurface aperture to changes in the ambient environment is mapped out. This technique produces target regions in which statistical analysis of charged particle accumulation will be considered most significant, allowing one to quantify the efficacy of the electrophoretic technique after simulation and experimentation.

Using HFSS (an electromagnetic simulation software), a 100 μm thick layer of FR4 (a circuit board material with $\epsilon^{\prime }=4.4$ and tanδ=0.02) with 100 μm diameter holes in it was digitally place on top of the metasurface, as shown in Figure 3.

Both holes were independently moved in the *m* and *n* directions from point *o* to point *p*, in nine equal steps. The transmittance spectra were simulated after each step from 90–110 GHz (the operational bandwidth of the metasurface). The maximum transmittance amplitude was recorded after each step. Once the transmittance amplitudes for the steps between *o* and *p* were recorded—due to the symmetry and periodicity of the metasurface—the values were mirrored across the origin. The resulting average maximum transmittance spectra after each step is presented in Figure 4.

Upon investigation of some of the features of Figure 4, it is interesting to note the enhanced sensitivity of the metasurface apertures (given by largest transmittance peaks) which correspond to the slits depicted in Figure 1, labeled *B*. This enhanced sensitivity in the region surrounding these slits is due to a massive electric field enhancement across them—a result of the resonant coupling of incident radiation to the metasurface topology, as detailed in [1]. Furthermore, the sensitivity of the metasurface abruptly drops to its least sensitive position at around 370 μm from the aperture centers. Upon closer examination, a major portion of the intensity distribution of the normalized transmittance spectra as a function of $D_{c}$ (the distance from the center of a metasurface aperture, as seen in Figure 4) can be very closely modeled using a modified Fraunhofer Diffraction model of a three-slit system [24]. The parameters of the modified Fraunhofer model are based upon the metasurface geometry, which is presented as Equation (Equation 1).
(1)$${\bar{S}}_{21}\left(x,y\right)=\alpha {\left(\frac{P}{2}\frac{k_{0}}{\sqrt{B_{ave}d}}\right)}^{2}{\left(4{\left(x,y\right)}^{2}-\sqrt{P}\right)}^{2}\cos^{2}\left(n\frac{P}{2}\frac{k_{0}}{\sqrt{B_{ave}d}}\left(x,y\right)\right){\mathrm{sinc}}^{2}\left(n\frac{P}{2}\frac{k_{0}}{\sqrt{B_{ave}d}}\left(x,y\right)\right)$$

In Equation (Equation 1), α is the normalization factor for the transmittance spectra, *P* is the period of the metasurface, $k_{0}=\frac{1}{\lambda_{0}}$ (the wavenumber at the resonant wavelength of the metasurface), $B_{ave}$ is the average slit width, and *d* is the inter-slit spacing. A three slit system is proposed because, in the m-n symmetries of the metasurface apertures, there are three slits of finite size, as seen in Figure 3. In the case of the m-symmetry, one may still refer to the central discontinuous slit, as a slit (and it seems to work as will be shown below). The full fit of this model to the simulated transmittance data which probes the sensitivity of the metasurface apertures (Figure 3) is presented in Figure 5.

As seen in Figure 5, there is a region in which the Fraunhofer model is most accurate: a region corresponding to one half of the period, *P*, from the center of each metasurface aperture (depicted in Figure 6). This could be a region of interest when considering how one may quantify the accumulation of charged particles around the metasurface apertures, considering that the metasurface is most sensitive to changes in its ambient environment within this region (evidenced by largest change in transmittance amplitude).

To further quantify the regions of greatest sensitivity, the spline fit, as shown in Figure 4, was integrated across the data set. If $S_{21}\left(x^{\prime }\right)$ represents the equation of the spline as a function of distance from the center, the values for $x^{\prime }$ for which the area under the spline contained 95% of the area, and the Fraunhofer region, were found and are presented in Table 2. The exact regions on the metasurface are depicted for a single aperture in Figure 7.

### 2.3. Electrostatic Fields through Metasurface Apertures

In this subsection, the fundamental physics which governs the nature of electric fields as they pass through small apertures will be examined.

In a simplified model of a metasurface aperture (a circular hole in the top of a parallel plate capacitor), Bansevicius and Virbalis calculated the electric field that could emanate from a circular hole in the positive plate of a capacitor [25]. To offer some insight into what an electric field looks like as it passes through an array of metasurface apertures, the electric field through a portion of the metasurface was simulated in Mathematica. A potential difference of 5 kV was setup between the ground plate and the metasurface (as depicted in Figure 2) using Dirichlet boundary conditions. Using a meshing technique, the ground plate and metasurface were subjected to finite element analysis. The bounding box surrounding the meshed regions was placed sufficiently far away in order to eliminate edge effects from the simulation, and the potential at those boundaries set to 0 V [26]. An aerial view of the surface which will be simulated is given in Figure 8, and the fields produced above the metasurface are presented in Figure 9.

The central unit cell will be the region of interest (region enclosed within black box of Figure 8) for statistical analysis within this paper. Although, as described in Section 2.2, the sensitivity of an individual unit cell aperture does not meaningfully extend beyond a distance of the metasurface period *P* from its center, the electrostatic fields which emanate from the apertures interact on much larger scales. Therefore, the central unit cell was surrounded by apertures in accordance to the periodicity of the metasurface such that, in the reference frame of the central unit cell, it may be embedded in a much larger structure. In order to more clearly depict the electric fields above the metasurface the density and contour plots were produced, and are presented in Figure 10.

Figure 10 show that the electric field which passes through the metasurface apertures extends well above the metasurface. Furthermore, one can see the contours of the electric field as they extend above the metasurface. Near the surface, the effect of the aperture geometry on the electric field is revealed by the contour lines.

In Figure 11, the gradient of the electric field within the central unit cell is depicted. As can be seen, there is a steep gradient toward the center of the metasurface (the point between the corners of the meta-atoms) and the large aperture slits, indicating that one should expect to see a relatively large accumulation of particles within this region. Furthermore, within the boundaries of the meta-atoms, one observes a double-lobe like structure which creates a steep gradient toward the largest aperture slits, also supporting this prediction.

### 2.4. Charged Particles Suspended in an Evaporating Medium While Subject to Nonuniform Electric Fields

In simulations and experiments presented herein, charged particles (colloidal latices with net positive surface charge density) are suspended in a nonuniform electric field which emanates from multiple metasurface apertures (see Figure 9). When an electric field (uniform or nonuniform) is applied to charged particles in suspension, they migrate along the field lines in a predictable manner such that a relative motion between the charged particles and the suspending medium is observed [27]. The electrostatic force on the charged particles is given as ${\vec{F}}_{E}$. Such charged particles in suspension will polarize the suspending solution surrounding it, giving rise to a counter-charged ion cloud called an electric double layer. Since the induced ion cloud is oppositely charged in comparison to the particle, the overall motion of the fluid (suspension medium) will be in the opposite direction of the charged particle’s.

The opposing suspension motion tends to retard the motion of the particle in suspension as an electric force is applied to it, creating a drag-like force called Electrophoretic Retardation (${\vec{F}}_{D}$). However, charged particles within the electrolyte are still able to move with a constant velocity, called the drift velocity ($v_{d}$). In such an instance, the charged particle-double layer volume can be treated as a single particle, with an effective radius of the moving particle including any molecules of water or other solvent that move with it, called Stoke’s Radius ($r_{S}$). The impact that the electric double layer has on the kinematics of the particle are determined by the layer thickness. The thickness of the electric double layer is represented by the Debye length $\kappa^{-1}$, and is a characteristic distance from the charged particle in the solution, to a radial distance from the surface normal of the charged particle in which the electric potential decays to approximately 33% of the total surface potential [27]. As a charged particle travels through a solution, there is a plane beyond which the oppositely charged ions do not travel along with the particle of interest: the slipping plane, which is defined by the electrical potential at the plane, and the zeta potential, which dictates the amount of repulsion between the surrounding counter-ion layer and the solution. Interestingly, this repulsion is due to the osmotic pressure created between the ionic concentration difference between the double layer and the solution [28,29].

However, when taking the Debye-Hückel condition, $\kappa^{-1}r>>1$ (where *r* is the Stoke’s Radius), and there is no pressure gradient within the suspension medium, there will be no flow of the suspension medium due to the electric field as there should be no other free charges present within it [27]. This means that the polarized electric double layer is considered to be very thin, and thus, the Electrophoretic Drag can be treated as negligible. The Debye-Hückel limiting case was considered in this paper, as the surface charge density of latices is considerably small (as will be seen in Section 2.7). This means that Stoke’s Radius can be treated as the particles radius. Furthermore, since the electric field that the particles are subjected to is non-uniform in this experiment (shown in Figure 9, Figure 10 and Figure 11), the electrostatic force contribution to particulate motion in the Debye-Hückel limit can be expressed as [27,30,31]:(2)$$F_{E}=qE$$
where *q* is the charge of the particle, and *E* is the electric field strength.

In a nonuniform electric field, an electrically neutral body in suspension will usually move in a direction in which the electric field gradient is the steepest; a phenomenon which was coined Dielectrophoresis by Pohl in 1951 [32]. The mechanism governing dielectrophoresis depends on the asymmetrical induction and attraction of charge densities within suspensoids, where electric dipoles are induced due to the charge asymmetry [32,33]. In simple dielectrophoresis (of an uncharged particle), the motion of the particle is dependent on the dielectric properties of the suspension medium and the particle.

In 1979, Pohl showed that the an effective dipole moment could be utilized in the formulation of the dielectrophoretic force [34]. For a perfectly insulating spherical particle of radius *r* with absolute permittivity $\epsilon_{2}$ that is suspended in a medium of absolute permittivity $\epsilon_{1}$, the dielectrophoretic force can be shown to be:(3)$$F_{Di}=2\pi r^{3}\frac{\epsilon_{1}\left(\epsilon_{2}-\epsilon_{1}\right)}{2\epsilon_{1}+\epsilon_{2}}\nabla {\left|E\right|}^{2},$$
where $\frac{\epsilon_{1}\left(\epsilon_{2}-\epsilon_{1}\right)}{2\epsilon_{1}+\epsilon_{2}}$ is the Clausis-Mosotti factor. In Equation (Equation 3), it is seen that the dielectrophoretic force is not explicitly dependent on any electrical charge within the medium or particles, but depends on the gradient of the nonuniform electric field and the dielectric properties of the medium and particles [30,32]. Upon further observation of Equation (Equation 3), one sees that particles are attracted to regions of stronger electric field when $\epsilon_{2}>\epsilon_{1}$, and repelled when $\epsilon_{2}<\epsilon_{1}$, and in general, is not parallel to the electric field [33,35,36].

When a solution is spread across the metasurface, it will naturally evaporate over time; in fact, this is a feature of this metasurface sensing technique that must be considered since the nature of latex sedimentation is highly dependent on evaporation and the properties of the deposition surface [20]. This phenomenon is not unique to latices or colloidal particles. For instance, in a NaCl solution (with sub-nanometer sized particles) evaporating on a hydrophilic surface, the majority of the salt crystallization during natural evaporation takes place at the liquid-air interface. This results in a ringlike crystal “coffee stain” effect, which is attributed to capillary flow from the center to the edges of the water droplet due to loss of the solvent and “pinning” of the water droplet to imperfections on the substrate surface. However, on a hydrophobic surface, the re-crystallization takes place within the bulk of the water droplet at the liquid-solid interface [17].

For hydrophobic colloidal particles (such as latices), it has been observed that they will preferentially adsorb onto hydrophobic regions, and in regions of opposite charge. However, it is known that latex films do not dry uniformly. Instead, due to the boundary conditions at a droplets edge, latex particles are laterally transported from the fluid centre to the edge of the droplet where they will be consolidated. This means that, in the case of no external forces but that of evaporation, a high concentration of latices should be observed along the perimeter of a water droplet [20].

In an ambient environment ($20^{{}^{\circ}}$ Celsius, and some air flow above the surface), it has been shown that the rate of evaporation of water can be approximated as within $\dot{m}=5\cdot 10^{-8}\frac{\mathrm{kg}}{s}$ [37]. In order to test the evaporation rate of the solution as it sat on the metasurface, five single drops of the colloidal solution used in this study were placed onto the metasurface and the time to evaporation measured. The ambient temperature was $24^{{}^{\circ}}$ Celsius, and the 12 V computer fan was turned on within the containment box. The fan was not directly pointing at the water droplets, but only allowed air to circulate within the containment box. It was found that, on average, it took approximately fifteen minutes for the water droplets to evaporate. Each of the droplets was measured to be approximately 3 mm in radius (radius parallel to the metasurface), with a height of approximately 1.5 mm. Modeling the droplet as half an oblate spheroid [38], the rate of evaporation for the water in our controlled environment was determined using the following equation:(4)$${\dot{m}}_{exp}=\frac{\frac{2}{3}\pi r_{exp}^{2}h\rho_{H_{2}O}}{t_{f}},$$
where $r_{exp}$ is the equatorial radius (radius of the sphere in the plane parallel to and on the metasurface), *h* is the polar radius (the height of the oblate spheroid) [38], $\rho_{H_{2}O}$ is the density of water, and $t_{f}$ is the approximate average time it took for a droplet to evaporate. Given that $r_{exp}=3$ mm, $\rho_{H_{2}O}=1000$ kg/m${}^{3}$, and $t_{f}=900$ s it was found that ${\dot{m}}_{exp}=3.14\cdot 10^{-8}$ kg/s.

Therefore, assuming that the evaporation rate of the water remains relatively constant, the change in volume of the water as a function of time can be described by the following equation:(5)$$\frac{dV}{dt}=-\frac{\dot{m}}{m_{0}}\left(V_{0}-V\right).$$

It can be shown that this separable differential equation results in an equation for the volume of the water on a surface as a function of time, V(t), where $m_{0}$ is the initial mass of the water on the surface, $V_{0}$ is the initial volume and $\dot{m}$ is the evaporation rate. Given that the estimated total time it takes to evaporate the water will be $\frac{1}{t_{f}}=\frac{m_{0}}{\dot{m}}$. Solving Equation (Equation 5) produces the following equation:(6)$$V\left(t\right)=A\cdot V_{0}\left(1-e^{\frac{t-t_{f}}{t_{f}}}\right),$$
where A is a normalization constant. Solving for *A* while considering that the volume must approach zero at $t_{f}$, one finds $A=\frac{e}{e-1}$. Keeping the experimentally measured evaporation time and geometry of the droplet of water in mind, the resulting equation is:(7)$$V\left(t\right)=\frac{e}{e-1}\cdot 2.827\cdot 10^{-8}\left(1-e^{1.111\cdot 10^{-3}\cdot t\;-\;1}\right).$$

A graph of the volume of the water on the metasurface as a function of time is given in Figure 12. In Figure 12, it is clear that the change in volume as a function of time is nonlinear, and therefore, the change in the radius and height of the water droplet should be nonlinear as well. Assuming that the radius of the water spot (approximated as a hemisphere) and its relative height from the surface should both reach zero at $t=t_{f}$, $\dot{r}\left(t\right)=\frac{r_{0}}{h_{0}}\dot{h}\left(t\right)$, such that in the event that $r_{0}=3$ mm and $h_{0}=1.5$ mm, the radius should decrease at a rate that is two times faster than the height of the droplet at a constant evaporation rate.

Equation (Equation 7) can be re-expressed in terms of the height of the water spot h(t) and the radius of the water spot r(t). Then differentiating each with respect to time, the rate at which the height and radius of the droplet can be determined. The graphs of r(t) and h(t) are given in Figure 13a,b. This phenomenon subjects the particles to another type of force which accelerates the sedimentation process, thus altering the equation of motion [39,40,41]. The exact rate of change for the radius ($\dot{r}$) and height ($\dot{h}$) of the water spot are given in Figure 13c and Figure 13d, respectively.

During conventional procedures of latex film deposition, it has been shown that evaporation is the driving force behind particle sedimentation. In such film deposition procedures, latex deposition is driven by the directed motion of particular influxing to evaporative regions via the convective mass transfer of water. As was previously described, the dominating sedimentation process takes place at the radial boundary of the water spot on top of the metasurface (due to convective mass transfer), directed in a downward direction at the edge due to the curvature of the water droplet. Since the total evaporation flux of the water must be equal to the bulk water flux to the boundary, it is suggested that the particle velocity due to the bulk water flux, $v_{⊥}\left(t\right)$, can be described as [39,40,41]:(8)$$v_{⊥}\left(t\right)=\dot{r}\left(t\right)\frac{A_{S}\left(t\right)}{A_{B}\left(t\right)},$$
where $\dot{r}\left(t\right)$ is the radial velocity due to evaporation as a function of time, depicted in Figure 13c, $A_{S}\left(t\right)$ is the total surface area of the water spot (half of an oblate spheroid), and $A_{B}\left(t\right)=4\pi r\left(t\right)r_{p}$ is the surface area of the radial boundary at which a two-dimensional lattice of particles has settled (estimated as the surface area of a cylinder whose height is the particle diameter), and $r_{p}$ is particle radius [39,40,41]. The surface area of an oblate spheroid is defined as [38]:(9)$$A_{s}\left(t\right)=2\pi r^{2}\left(t\right)-\pi \frac{h^{2}\left(t\right)}{e}\ln\frac{1+E}{1-E},$$
where r(t) is the equatorial radius (radius of the sphere in the plane parallel to and on the metasurface), h(t) is the polar radius (the height of the oblate spheroid), and $E=\sqrt{1-\frac{h^{2}\left(t\right)}{r^{2}\left(t\right)}}$. Solving Equation (Equation 8), one will find that:(10)$$v_{⊥}\left(t\right)=0.31\frac{\dot{r}\left(t\right)r\left(t\right)}{r_{p}}$$

It is suggested here that, since the latex particles within the medium are subject to the convective mass transfer of water due to evaporation, using Stoke’s law [42], the downward force contribution that the evaporative process contributes to particle motion within the solution can be described by:(11)$$F_{S}=6\pi \eta r_{p}v_{⊥}\left(t\right)+\frac{4}{3}\pi g\left(\rho_{p}-\rho_{f}\right)r^{3},$$
where $r_{p}$ is the particle radius, η is the fluid viscosity, $v_{f}$ is the particle velocity due to fluid flux, *g* is the acceleration due to gravity, $\rho_{p}$ is the particle density, and $\rho_{f}$ is the fluid density. Furthermore, the contribution from frictional drag due to Stoke’s Law will retard the motion of the particles in directions which are not parallel to the convective mass transfer vectors. This means that, since the majority of the latice deposition takes place at the evaporative front, the force vector due to evaporation ($F_{S}$) contributes to the downward motion, whereas motion parallel to the metasurface is retarded in a manner predicted by Stoke’s Law. The retarding force due to frictional drag, whose vector components are parallel to the metasurface, are given by: (12)$$F_{S}=6\pi \eta r_{p}v_{\|}\left(t\right).$$

Finally, the net force on a charged particle in suspension subject to an nonuniform electric field during evaporation can be expressed as:(13)$$\Sigma \vec{F}={\vec{F}}_{E}+{\vec{F}}_{Di}+{\vec{F}}_{S}=q\vec{E}+2\pi r_{p}^{3}\frac{\epsilon_{1}\left(\epsilon_{2}-\epsilon_{1}\right)}{2\epsilon_{1}+\epsilon_{2}}\nabla {\left|\vec{E}\right|}^{2}+6\pi \eta r_{p}\left({\vec{v}}_{⊥}+{\vec{v}}_{\|}\right),$$
where downward acceleration due to gravity is negligible within the fluid. In the limit that the electric field goes to zero, evaporation will be the only external driving force which causes particle deposition onto the surface.

Due to the convective mass flow of the water toward the edges of a water droplet, it should be observed experimentally that the particles should aggregate in clumps near the edge, where a random, smaller distribution of particles is observed over the wider area [20] without the presence of an externally applied electric field. Conversely, in the presence of an electric field, it should be that the latices are not so highly concentrated along the boundary of a water droplet, but instead, their bulk should become increasingly spread out over a larger surface area and concentrated near metasurface apertures in a manner proportional to the electric field strength they are subjected to.

### 2.5. Monte Carlo Simulations of Charged Particles Suspended in an Evaporating Medium While Subject to Nonuniform Electric Fields

In order to probe the efficacy of the electrophoretic technique which is the focus of this paper, physical parameters were used in order to accurately approximate realistic effects that may be observed in experiment.

Spherical, colloidal latex particles were simulated using this Monte Carlo technique and given radii of r = 250 nm, 500 nm, 1000 nm, and 2500 nm. This range of radii for various latices is well documented [39]. From these initial conditions, the mass of each particle (*m*) and its net charge (*q*) can be readily calculated using existing data. Table 3 shows the parameters which were used in the Monte Carlo simulations, where m = ρ·V is the total mass of each of the colloidal spheres and *q* is their net charge. The average surface charge density on a number of latex particles has been previously quantified, and is given as $\sigma_{L}$ [43], the density of industrialized (non-natural) latex is given as approximately 1100 $\frac{\mathrm{kg}}{m^{3}}$ [44], and the relative permittivity of latex given as $\epsilon_{l}=2.5$ [45]. Furthermore, since the electric field between the metasurface and ground plate travels through a 400 μm silicon wafer ($\epsilon_{S}=11.9$), a 0.8 mm thick sheet of Teflon ($\epsilon_{T}=2.1$) [46], and then effects particles that are within water as a suspending medium ($\epsilon_{H_{2}O}=80$) [47], the electric field in each of these regions was diminished by the relative permittivity within them, respectively.

In this set of Monte Carlo simulations, particles (whose physical parameters are given in Table 3) were given an initial position within a 2.9 mm × 2.9 mm × 1.5 mm volume centered directly on top of the metasurface region depicted in Figure 8. One by one, 10,000 particles bearing the physical parameters presented in Table 3 were exposed to an electric field above the metasurface which was created by a potential difference between the metasurface and the ground plate. Upon using a Monte Carlo technique, it is assumed that the particle concentration within the suspending medium during experimentation will be sufficiently small, such that the particles can be considered non-interacting within the medium. The voltage range explored in these simulations varies from 0–15 kV, with simulations of the 250 nm particles reaching 50 kV (why this is so will be later explained). The net force on each of the particles is given by Equation (Equation 13). The efficacy of the technique was quantified using the Fraunhofer Accurate and 95% regions of sensitivity (within the central unit cell shown in Figure 8) previously quantified in Section 2.2, and depicted in Figure 7. The justification to study only the central unit cell was previously given in Section 2.3.

The gathering efficiency within the Fraunhofer Accurate and 95% Sensitivity Regions is defined as $\frac{N_{r}}{N_{T}}$, where $N_{r}$ is the number of particles found in each of the regions of interest, and $N_{T}$ the total number of particles studied. The initial gathering efficiency of the metasurface within the Fraunhofer Accurate and 95% Sensitivity Regions was quantified by setting the electric field to zero in the simulation, and determining the number of particles that accumulated within each of the sensitivity regions after it completed. The sedimentation of the particles purely due to evaporation is presented in Figure A1, and the initial gathering efficiency within these regions presented in Table 4.

The gathering efficiencies of the metasurface as a function of voltage between the metasurface and grounding plate ($\epsilon_{r}\left(V\right)$) within this study are well modeled utilizing the equation:(14)$$\epsilon_{r}\left(V\right)=\left(\epsilon_{m,r}-\epsilon_{V_{0},r}\right)\frac{V}{V+V(\frac{\epsilon_{m,r}+\epsilon_{V_{0},r}}{2})}+\epsilon_{V_{0},r}$$
where, $\epsilon_{m,r}$ is the maximum efficiency possible within the metasurface region of interest (in the limit that V→∞), $\epsilon_{V_{0},r}$ is efficiency within the region of interest when V=0 (given in Table 4), and $V(\frac{\epsilon_{m,r}+\epsilon_{V_{0},r}}{2})$ is the voltage at which one observes one-half the sum of the maximum and minimum efficiencies within the region of interest. Furthermore, the shape of the increase in efficiency as a function of voltage (which will be presented in Section 2.6) suggests that Equation (Equation 14) is a member of a class of rectangular hyperbolic growth models. According to Srinivasan and Rao [48,49,50],

A rectangular hyperbola is a curve that describes a plethora of biological processes including dependence of reaction velocity on substrate concentration in biochemistry, maximum effect of drug concentration in pharmacology, photosynthetic rate/growth rate on light intensity/nutrient level in physiology, microbial growth rates in aqueous environments to the concentration of a limiting nutrient (Monod equation), type II functional response as emphasized by Holling’s disc equation showing the relationship between prey density and prey consumption by predator, several instances in population ecology, Langmuir isotherms and many more.

One such physical process which is extremely similar to the gathering efficiency of the metasurface presented in Equation (Equation 14) is governed by the Michaelis-Menten equation, which is one of the best-known models of enzyme kinetics, given by $v=V_{m}\frac{\left[S\right]}{\left[S\right]+k_{m}}$, where [S] is the concentration, $V_{m}$ is the maximum reaction rate achievable by the system (in the limit that [S]→∞), and $k_{m}$ is a constant and the concentration at which one achieves half of the maximum reaction rate possible ($\left[S\right]\left(\frac{V_{m}}{2}\right)$) [50]. In fact, the main difference between Equation (Equation 14) and the Michaelis-Menten equation is that Equation (Equation 14) considers a non-zero initial efficiency. To show that one recovers a Michaelis-Menten-like equation in the event of a starting efficiency of zero, one may take:(15)$$\lim_{\epsilon_{V_{0,r}}\to 0}\epsilon_{r}\left(V\right)=\epsilon_{m,r}\frac{V}{V+V(\frac{\epsilon_{m,r}}{2})}.$$

### 2.6. Gathering Efficiency of $r_{p}$ = 250 nm, 500 nm, 1000 nm, and 2500 nm Particles Estimated via a Monte Carlo Method

The efficiency of the electrophoretic technique was theoretically quantified using the same procedure described at the beginning of Section 2.5, where the gathering efficiencies of the metasurface as a function of voltage were identically modeled after Equation (Equation 14). The efficiency of the electrophoretic technique ($\epsilon_{r}\left(V\right)$) as a function of potential difference between the metasurface and the grounding plate within the Fraunhofer ($\epsilon_{F^{*}}\left(V\right)$) and 95% ($\epsilon_{95}\left(V\right)$) regions of sensitivity for particles of each radii are presented in Figure 14 and Figure 15, respectively. Using nonlinear regression, fits taking the form of Equation (Equation 14) for each of the data sets were found. The equations of best fit, and which govern the gathering efficiency as a function of voltage within the Fraunhofer Accurate and 95% Sensitivity Regions, are given in Table 5. Furthermore, $\epsilon_{m,r}$ were determined by taking the limit as V→∞ for $\epsilon_{r}\left(V\right)$, and are also provided in Table 5.

Figure 14, in conjunction with Table 5, shows that the rate at which each of the particle gathering efficiencies approaches their asymptotic limit is proportional to their radii (and thus, surface charge) within the Fraunhofer Accurate Sensitivity Region. This is largely due to the fact that the surface charge for each of the particles tested differs from one another by an order of magnitude, suggesting that the larger, more highly charged particles should experience a much larger force than their smaller counterparts. In contrast to this observation, the theoretical maximum efficiency achievable for each particle of radius $r_{p}$ is inversely proportional to their radius, suggesting that much higher voltages may be necessary to elicit the desired effect (if the net charge of the particles scales in such a nonlinear fashion such as in the case of the latex colloids).

In a similar manner to Figure 14, Figure 15 shows the same behavior in the gathering efficiencies as a function of voltage and particle radius within the 95% Sensitivity Region. However, in the case of the 2500 nm radius particles, the fit equation more accurately depicts the gathering efficiency. This will be an important consideration in further analysis below. In Figure 14 and Figure 15, it is known that in the case of the 250 nm radius particle that the fit curves are accurate because the Monte Carlo simulation was further run for 20 kV, 30 kV, and 50 kV. Within this range, of voltages tested (0–50 kV), the gathering efficiency very closely matches the equations listed in Table 5 for the Fraunhofer Accurate and 95% Sensitivity regions for the 250 nm particles, as seen in Figure A2.

The average difference between the number of particles within the Fraunhofer Accurate and 95% Sensitivity Regions does not exceed 4% for any particle radii tested. This suggests that, of all the particles that settled within the much larger 95% Sensitivity Region, the majority of them are also within the Fraunhofer Accurate sensitivity region. This is an advantageous feature of this technique since, as depicted in Figure 5, the greatest sensitivity of the metasurface lies within the Fraunhofer Accurate region. Figure 16 is provided as evidence of the small difference in gathering efficiency between the Fraunhofer Accurate and 95% Sensitivity Regions. Furthermore, as an example of how $\epsilon_{m,r}$, $V(\frac{\epsilon_{m,r}+\epsilon_{V_{0},r}}{2})$, and $\epsilon_{V_{0},r}$ from Equation (Equation 14) relate to each of the data sets, Figure 17 is provided.

The relative percent increase in gathering efficiency as a function of voltage is given as $\varepsilon_{r}=\frac{\epsilon_{r}\left(V\right)-\epsilon_{r}\left(V_{0}\right)}{\epsilon_{r}\left(V_{0}\right)}$. The relative increase in gathering efficiency within the Fraunhofer Accurate and 95% Sensitivity Regions is presented in Figure A3 and Figure A4, and their respective equations in Table A1. Within the Fraunhofer Accurate region, there is approximately a 1000%, 1400%, 1600%, and 1600% increase in the efficacy of the electrophoretic technique at 15 kV when compared to random sedimentation due to evaporation (when V = 0) for the $r_{p}=250$ nm, 500 nm, 1000 nm, and 2500 nm particles, respectively. Within the 95% Sensitivity region, there is approximately a 800%, 1000%, 1200%, and 1200% increase at 15 kV for the $r_{p}=250$ nm, 500 nm, 1000 nm, and 2500 nm particles, respectively.

As a graphic example of the gathering efficiency as a function of voltage using this electrophoretic technique, Figure 18 is provided for the reader, which depicts the gathering efficiency within the area presented in Figure 8 for the 500 μm radius latices described in Table 3. As can be seen in Figure 18, the particles are forced to settle closer to the center of the entire area presented in Figure 8 as the voltage increases. Some interesting and typical features observed at many voltages after simulating the electrophoretic technique are presented in Figure 19. If one looks closely at Figure 18, they will see similar features at other voltages not depicted in Figure 19. In Figure 19, the blue oval in the 10 kV frame shows what appear to be tendril-like extensions from the largest slits in the metasurface aperture. These tendrils appear to be stretching out to meet the center unit cell apertures. The green circle in Figure 19 encloses a line of particles which runs from the bottom left corner of the central most unit cell, diagonally toward the center of the frame. At the center of the black oval in Figure 19, the particles have extended from corner to corner of the central most unit cells in the frame. The density of the particles at 15 kV is well represented in Figure 20.

Figure 20 is a three dimensional histogram, focused on the center-most unit cell of the 15 kV frame from Figure 19. The histogram bins are 60 μm × 60 μm (2500 bins total on the surface). According to this histogram, there appears to be a very high concentration of particles at the corners of the metasurface atoms, at the center of the frame. Furthermore, between the corners of the metasurface atoms, a less populated mass extends between them, bridging the gap. Figure 19 and Figure 20 hint at the types of densities and structures that may be observed experimentally. However, it was suspected that since the particles would like to minimize their potential energy on the surface, instead of stacking upon one another in a never ending fashion during the experiment, they will be forced to spread out as much as the electric field allows them to near the highest density regions depicted in Figure 20. This may lead to a more evenly distributed “bridge” in the density of the particles between the meta-atom corners.

### 2.7. Experimental Parameters

To test the electrophoretic technique, an experimental apparatus was constructed, whose diagram is presented in Figure 2, and image is shown in Figure 21. A Glassman High Voltage power supply was connected to the metasurface and ground plate via high voltage wire. The metasurface (manufactured at the University of Massachusetts Lowell Emerging Technologies and Innovation Center’s Nanofabrication Laboratory using photolithography), ground plate and fan were encased in an acrylic containment box with a 12 V computer fan placed inside to accelerate evaporation (facing at the corner, away from the metasurface).

A 15 mL solution of 10% by mass, aminated 500 nm radius spherical latex nanoparticles suspended in distilled water was obtained from MAGSPHERE, INC. (Pasadena, CA, USA). The specific gravity of the particles were estimated by MAGSPHERE, INC. to be approximately 1050 $\frac{kg}{L}$. No exact surface charge information was provided by MAGSPHERE, INC., and we lack the ability to precisely measure such properties. 0.5 mL of the 10% latex nanoparticle solution was diluted in 150 mL of distilled water, producing a solution that was approximately 333 parts per million (ppm) latex nanoparticles, by mass. The 333 ppm latex nanoparticle solution was deposited (at room temperature) onto the metasurface a single drop at a time using a serological pipette. After deposition of a single drop, the containment lid was closed, the apparatus turned on, and the droplet left to evaporate. In total, seventeen drops were placed onto the metasurface. Each drop was independently exposed to a successively increasing electric field strength via the potential difference between the metasurface and the ground plate. For instance, the first drop was left to evaporate under 0 V between the metasurface and ground plate, the second drop was left to evaporate under 250 V, and the third through the seventeenth drops under 1–15 kV (increasing by 1 kV increments from the third drop up to the seventeenth drop).

Since it was suspected that the particles could not be considered non-interacting at a concentration of 333 ppm (based upon visual inspection of the droplet area after evaporation), a second test of the experiment was conducted in order to more accurately test the Monte Carlo model presented in Section 2.5. To do so, 0.25 mL of the 10%, 500 nm radius latex nanoparticle solution was diluted in 200 mL of distilled water, producing a solution that was 125 ppm latex nanoparticles by mass. After the 333 ppm and 125 ppm droplets evaporated, the resulting particle distributions were imaged via an inspection scope and a high powered microscope. The resulting particle distributions were then compared to the morphological results obtained via the Monte Carlo simulations.

After inspecting the efficacy of the electrophoretic technique using the 333 ppm and 125 ppm solution, the metasurface was thoroughly cleansed of latices using pure Acetone and Class 100 Cleanroom Wipes (confirmed under microscope and inspection scope), and the entirety of the sensing surface covered with 5 mL of the 333 ppm solution. As observed in [1], the metasurface utilized in this study has a resonant wavelength of approximately 2.8 mm, and may be capable of detecting inhomogeneous powder layers on its surface which are effectively smaller than 20 μm thick. However, to be sure that we would have the greatest chance of detecting a change in the resonant spectra of the metasurface in the presence of the 500 μm radius latex nanoparticles at W-band, the 333 ppm solution was used. First, 5 mL of solution was deposited on the entire sensing area of the metasurface and left to evaporate without any electric field applied to it. Using the experimental apparatus depicted in Figure A5 [1], the transmittance spectra of the metasurface with latex nanoparticles deposited onto its surface, measured between 90–110 GHz (with a frequency resolution of 16 MHz and dynamic range of 80 dB), was recorded using a Network Analyzer and set of W-band lenses and horns, following the same measurement procedure stated in [1]:

The Network Analyzer was calibrated using a response calibration technique. Only transmittance data, denoted in this study as $S_{21}$ (the total transmitted power detected by Horn 2 as it leaves Horn 1), was considered. The horns were erected approximately 80 cm apart. The Gaussian beam produced through the focusing lenses at each horn had a FWHM of approximately two inches in diameter in the focal plane and was centered on the metasurface (4-inch diameter). The measurement window was isolated from the ambient environment and the effects of noise were removed from the calibrated data by applying time-domain gating of the space outside of the sample area. Furthermore, the optical bench that the apparatus was erected on, and various metallic surfaces in the area, were covered in mm-wave absorbing material to further reduce error in the measurements.

Then the metasurface was thoroughly cleaned again, and the 333 ppm solution was reapplied to the entire sensing area of the metasurface. It was then left to dry under 15 kV between the metasurface and the ground plate. After evaporation, the transmittance spectrum of 333 ppm solution which dried under 15 kV was measured. The 15 kV 5 mL spot test was conducted three separate times. The spectra of the 0 V and the three 15 kV latex films was then compared to the spectra of the bare metasurface. The experimentally measured transmittance spectrum of the bare metasurface is presented in Figure 22.

## 3. Results

### 3.1. Gathering Efficiency as a Function of Voltage within Droplet Regions

In order to determine the efficacy of the electrophoretic technique as a function of voltage using the methodology described in Section 2.7, each of the droplets were imaged via an inspection scope and high powered microscope. An inspection scope was used in conjunction with a high powered microscope because the field of view is much greater than the microscope is capable of providing. Images of a metasurface region with no droplet applied to it, and a droplet area after evaporation with no potential difference between the metasurface and grounding plate are presented in Figure 23A and Figure 23B, respectively.

Each of the spots imaged via the inspection scope were illuminated with white light, at a constant brightness. In Figure 23A, the metasurface apertures are faintly visible (no latex solution), and in Figure 23B, anything that is white within the black circular region is where latex particles have settled after evaporation. As predicted in Section 2.4, it appears that the majority of the particles within the evaporative region of the droplet (Figure 23B) gathered around the perimeter at 0 V, with a large dominating clump at the bottom right of the image. The exact cause of the dark red nature of the droplet area depicted in Figure 23B is unknown. However, it is suggested that the dark red reflection could be caused by the scattering of the source radiation (white light) due to the latice spheres which are spread across the metasurface (and the small spaces between the spheres). Furthermore, since the inspection scope camera is several inches away from the metasurface while imaging the largest field it is capable of (this is true for Figure 23), perhaps the red wavelengths are insufficiently scattered at this distance to not be detected by the camera.

The droplets which were exposed to the electrophoretic technique, and imaged using the inspection scope are presented in Figure 24. The exposure settings and magnification of the inspection scope were identical to those used in the case of the 0 V droplet shown in Figure 23B, and each circular region represents the entirety of droplet area observed at each voltage. Looking at Figure 24, any of the bright white features within the droplet area are the latex nanoparticles.

The effect of the electrophoretic technique on the nanoparticles becomes more apparent at the highest voltages tested. Though, the structures which form do not lend themselves to prediction based upon successive increases in voltage, as the morphology of the particle accumulations vary drastically between applied voltages at this macroscopic scale [20]. However, from 12–15 kV, prominent tendril-like features begin to emerge en masse, extending well across the central regions of the droplet areas. These tendril-like features may be indicative of large quantities of particles becoming trapped by the electric field emanating from the apertures before they can reach the edge of the droplet via convective mass transfer. Within the 15 kV droplet area, it appears that there has been some unexpected electrical effect evidenced by the thin, densely packed particle vein which extends from the bottom of the droplet toward the center, and at which point is spread out almost symmetrically to the left and right side of the droplet area. The metasurface was not damaged in any way after the 15 kV test, and we found no evidence of electrical breakdown within the insulative materials used in this study. The exact nature of these tendril-like features are more clearly explored and represented in subsequent images in this section.

Figure 25 depicts some representative images taken via a high powered microscope (5× magnification) within the droplets at 0 V, 5 kV, 10 kV, and 15 kV. In these frames, the latex particles are the black clumps, dots, and lines on top of the metasurface. As can be seen within the 0 V droplet in Figure 25, there was very little accumulation of particles anywhere outside of the main clump depicted at the bottom of Figure 23B. This indicates that the metasurface apertures (250 nm deep) have little to no influence on the accumulation patterns of the latices with no electric field present. The 5 kV, 10 kV, and 15 kV images of Figure 25 represent some interesting structures which were observed in regions not directly connected to a major mass of particles, but which formed beyond them. The black, blue, and green arrows within Figure 25 point out a few features (not all) which are reminiscent of those enclosed by the black, blue, and green circles/ovals presented in Figure 19, respectively.

Due to the suspected appreciable interaction between particles within the 333 ppm solution, an attempt was made to reduce the impact of inter-particular interaction by reducing the concentration. A 125 ppm droplet of the latex nanoparticle solution was deposited onto the metasurface and left to evaporate at 15 kV in order to more closely test the conditions set forth in the Monte Carlo simulations. Figure 26 shows some interesting features observed at 5×, 10×, 20×, and 50× magnification within the 125 ppm droplet region after evaporation. Just as in Figure 25, the black spots represent particle accumulation, and the arrow color corresponds to regions enclosed in Figure 19.

Within the 5× image of Figure 26, one can see particle clumps one the surface. In the 10× image of Figure 26, an accumulation between the corners of the meta-atoms is observed, and the tendril-like extension which points to the right. In the 20× image of Figure 26, one can see a tendril-like structure which extends between the faces of two meta-atoms. The 50× image of Figure 26 is a further magnified region of the 20× region, but shows the actual packing behavior of the discreet particles within the aperture slits, and across the meta-atoms. In all, the 125 ppm solution appears to have produced structures which differ from those observed in 333 ppm evaporative regions. However, it is not clear that the Monte Carlo simulations have successfully modeled the experimentally observed phenomena, even with a decrease in the solution concentration.

### 3.2. The Impact of the Electrophoretic Technique on Metasurface Sensing Capabilities

In order to determine whether the electrophoretic technique could improve the sensing capabilities of the metasurface at W-band, 5 mL of the 333 ppm solution was applied to the functional area of the metasurface and left to evaporate at 0 V, and 15 kV in four separate runs of the experiment. Figure 27a depicts the spot on the metasurface in which 5 mL of the 333 ppm solution was deposited and evaporated at 0 V. Figure 27b–d depict the spots on the metasurface in which 5 mL of the 333 ppm was deposited and evaporated at 15 kV. In Figure 27, the spot diameters are approximately three inches wide, where the dashed black circle represents the spot size of the W-band radiation used in an attempt to sense the latices (approximately two inches in diameter [1]).

As can be seen in Figure 27, there are substantial differences between the distribution of particles with and without an external electric field present during evaporation. In Figure 27a, the vast majority of particles seemed to have accumulated in a large clump near the bottom of the metasurface (in accordance with theory); although, the clump notably extends within the W-band measurement region. This formation is most likely due to the constant flow of air from some direction due to the 12 V computer fan. Conversely, in Figure 27b–d, much more interesting structures are seen, such as strands of latices extending from left to right on the metasurface, and long tendril-like appendages branching out to different regions. Furthermore, it is clear that upon initial visual inspection that much more of the latices were able to be deposited across the larger area of the metausurface in Figure 27b–d than what was observed in Figure 27a, indicating that the electrophoretic technique is influencing the gathering efficiency of the metasurface. It should be noted, that upon cleaning the metasurface with the class 100 clean room wipes and pure acetone, that some of the unit cells became damaged. They damaged cells present themselves as the barren circular spots observed in Figure 27a–d.

To explore some of the typical types of structures which were observed in Figure 27a–d in detail, Figure 28 is offered. These images represent some of the interesting features that were observed using an inspection scope. Figure 28(a1–a3) are those images which represent the latice deposition at 0 V (from Figure 27a), and Figure 28(b1–b3) (from Figure 27b), Figure 28(c1–c3) (from Figure 27c), and Figure 28(d1–d3) (from Figure 27d), those seen at 15 kV. Comparing Figure 28(a1) to Figure 28(b1,c1,d1), Figure 28(a2) to Figure 28(b2,c2,d2), and Figure 28(a3) to Figure 28(b3,c3,d3), it is clear that there were major difference in the distribution of the latices with and without electrophoresis. The prominent differences are most clearly seen upon comparison of Figure 28(a2) to Figure 28(b2,c2,d2), and Figure 28(a3) to Figure 28(b3,c3,d3). In Figure 28(a2), the deposition of the latices does not seem to be related to the metasurface topology, which coincides with observations within the small droplet region previously presented. However, in Figure 28(b2,c2,d2), one can see long strands of latices which extend across the metasurface, seemingly interacting with the apertures in a manner not observed at 0 V. In Figure 28(a3), it is also clear that the latex strands that were created under evaporation at 0 V are not appreciably influenced by the physical structure of the metasurface apertures. In Figure 28(b3,c3,d3), however, it is clear that the structure of the long latex strands which are apparent in Figure 27b–d, are highly influenced by the electric field which is made to pass through the apertures. Although the latices appeared to have accumulated near the apertures in some regions of the metasurface, further investigation is necessary. The latice structures observed in experiment are different than those observed via Monte Carlo simulation, and the consistency of their accumulation is unpredictable.

To measure the effect that the electrophoretic technique has on the sensing capability of the metasurface, the transmittance spectra of the of the metasurface without any particles, with particles evaporated at 0 V, and with particles evaporated at 15 kV (three different experimental runs) was measured using the procedure detailed at the end of Section 2.7. This serves as a proof of concept of the proposed electrophoretic technique, as this paper is not a detailed analysis of material properties and their method of extraction, such as was presented in [1]. In [1], the technique developed for metasurface sensing is concerned with the maximum transmittance amplitude and the resonant frequency of the acquired spectra, as this allows one to visualize and quantify the spectral differences most easily at W-band. Figure 29 depicts the experimentally acquired maximum transmittance amplitude and resonant frequency of the spectra measured at W-band. The relative increase in the shifts in resonant frequency and transmittance amplitude (from the bare metasurface) at 15 kV (Figure 27b–d), over that observed at 0 V (Figure 27a) are presented in Table 6.

Although the shift in transmittance amplitude and resonant frequencies are small due to the presence of the latices, it is to be expected when dealing with an approximately 2.8 mm wave. In [1], the resonant frequencies and transmittance amplitudes of the thinnest inhomogeneous powder layers tested were also considerably small. However, given nearly 80 dB of dynamic range with a 16.67 MHz frequency resolution, the relative change in maximum transmittance amplitude and the resonant frequency due to the latices on the metasurface are substantial enough to quantify the difference between them (Figure 29). At 15 kV, the metasurface was able to register shifts in maximum transmittance amplitude and resonant frequency from the bare antenna resonance (purple square) which were approximately 900–1300% greater in resonant frequency, and 1408–1699% greater in maximum transmittance amplitude than those shifts observed at 0 V.

## 4. Discussion

In an attempt to generalize Equation (Equation 14) for a variety of particle types, $\left(\epsilon_{m,r}-\epsilon_{V_{0},r}\right)$ and $V(\frac{\epsilon_{m,r}+\epsilon_{V_{0},r}}{2})$ (from hereon, notated $\Delta \left[\epsilon_{m,r},\epsilon_{V_{0},r}\right]\left(r_{p}\right)$ and $k(r_{p})$, respectively) were fit as a function of particle radius. It was determined that the gathering efficiency of the electrophoretic technique can be closely represented as:(16)$$\epsilon_{r}\left(r_{p},V\right)=\Delta \left[\epsilon_{m,r},\epsilon_{V_{0},r}\right]\left(r_{p}\right)\frac{V}{V+k(r_{p})}+\epsilon_{V_{0},r},$$
where $r_{p}$ is the particle radius, $\epsilon_{m,r}$ is the maximum efficiency within the asymptotic limit of a particular region, and *V* the applied voltage. It was found that $\Delta \left[\epsilon_{m,r},\epsilon_{V_{0},r}\right]\left(r_{p}\right)$ and $k(r_{p})$ can be expressed in terms of the equations founds in Table 7 within the Fraunhofer Accurate and 95% Sensitivity Regions. The fits are displayed in Figure 30 and Figure 31.

It is suggested that there may be a relationship between the theoretical maximum efficiency and metasurface geometry. If one could relate the geometric parameters of the metasurface to the maximum expected efficiency given some particle radius (or perhaps some other particle property), perhaps a metasurface could be optimized for particular particles of interest. For example, within the 95% sensitivity region of the metasurface, one can estimate $\Delta \left[\epsilon_{m,95},\epsilon_{V_{0},95}\right]\left(r_{p}\right)$ as: (17)$$\Delta \left[\epsilon_{m,95,}\epsilon_{V_{0},95}\right]\left(r_{p}\right)=\frac{\int_{-x\left(\frac{m}{2}\right)}^{x\left(\frac{m}{2}\right)}{\bar{S}}_{21}\left(x^{\prime }\right)dx^{\prime }e^{-\frac{k_{0}\cdot g^{2}}{\sqrt{B_{ave}d}}r_{p}}+\int_{-x(m)}^{x(m)}{\bar{S}}_{21}\left(x^{\prime }\right)dx^{\prime }}{\int_{-\frac{p}{2}}^{\frac{p}{2}}{\bar{S}}_{21}\left(x^{\prime }\right)dx^{\prime }}$$where *g* is the average inter-patch spacing ($g=\frac{P}{\sqrt{2}}$, as detailed in [1]), $B_{ave}$ is the average slit width, *d* is the distance between slits, $k_{0}$ is the wave number at resonance, and each integral refers to the area under ${\bar{S}}_{21}$ depicted in Figure 5, given as Equation (Equation 1). Each of the integrals in the numerator are normalized by the area under the curve within the Fraunhofer Accurate region. The bounds of the first integral, given by ±x(0.5·m) are the distances from the aperture centers where one observes one-half the maximum sensitivity (which is one) in Figure 4. The bounds of the second integral, given by ±x(m), are the distances from the aperture center to where the maximum sensitivity is observed in Figure 5. The comparison of Equation (Equation 17) to that of $\Delta \left[\epsilon_{m,95},\epsilon_{V_{0},95}\right]\left(r_{p}\right)$ in Figure 30 is presented in Figure 32 The bounds of integration in relation to the Fraunhofer fit within the Fraunhofer Accurate Sensitivity Region are graphically presented in Figure A6.

The offered relationship between the geometry of the metasurface and the maximum gathering efficiency of the metasurface (Equation (Equation 17)) further supports the findings presented in Figure 16, such that the maximum gathering efficiency within the 95% Sensitivity region (nearly the entire span of the sensing capability of the metasurface) is almost entirely influenced by the Fraunhofer Accurate region of the metasurface. Furthermore, this is evidenced by $\frac{k_{0}}{\sqrt{B_{ave}d}}$ in the exponential of Equation (Equation 17), which also appears in the cos and sinc terms of Equation (Equation 1). However, it is suggested that the relationship between the geometric structure of a metasurface and its gathering efficiency be rigorously explored.

## 5. Conclusions

In this paper, a novel electrophoretic technique to improve metasurface sensing of low concentration, charged particles in solution has been presented. The particles of interest in this study were colloidal latex nanoparticles. A technique was proposed which allows one to quantify the metasurface regions of greatest sensitivity, and an equation proposed which very accurately models the sensitivity of the device within a distance from the aperture centers called the Fraunhofer Accurate Sensitivity Region (or one-half the period from the center). These regions of greatest sensitivity served as the regions upon which the efficacy of the electrophoretic technique was determined. The nature of electrostatic fields passing through metasurface apertures was explored (Figure 9, Figure 10 and Figure 11), and the equations which govern the motion of charged particles suspended in an evaporating medium while subject to nonuniform electric fields were presented (Equation (Equation 13)). It was suggested that, due to the gradient of the electric field which emanates through the apertures, one should observe a higher concentration of charged particles within the Fraunhofer Accurate Sensitivity Region in comparison to sedimentation under evaporation alone.

Monte Carlo simulations of the charged particle which were governed by said equations of motion were carried out for 10,000, 250 nm, 500 nm, 1000 nm, and 2500 nm radius latex particles (Table 3) suspended within distilled water from, 0–15 kV. The gathering efficiency as a function of voltage between the metasurface and grounding plate, or the ability for the metasurface to gather particles within the regions of greatest sensitivity was calculated and plotted after the Monte Carlo simulations for each of the particle radii tested (Figure 14 and Figure 15). A novel and general equation which very accurately describes the gathering efficiency of the metasurface was then presented (Equation (Equation 14)), and the exact equations which govern the increase in the gathering efficiency of the metasurface regions of greatest sensitivity as a function of voltage for each particle radius were presented (Table 5). Then, the behavior of the particles simulated using the Monte Carlo technique was presented (Figure 18), and some interesting latice structures formed due to the electrophoretic technique identified (Figure 19 and Figure 20). It was found that there is a clear increase in the gathering efficiency of the metasurface, as a function of voltage, while simulating the electrophoretic technique. The theoretical maximum relative increase in gathering efficiencies, and the governing equations for the various particle radii tested were then presented in Table A1. The maximum relative increase in gathering efficiencies from the Monte Carlo simulations ranged from an increase of 1554–1892% within the Fraunhofer Accurate Sensitivity Region, to 1172–1403% within the 95% Sensitivity Region.

An apparatus was constructed (Figure 2) in order to test the electrophoretic technique on 500 μm latices in solution. Latex nanoparticle solutions which were 333 ppm and 125 ppm by mass were dropped onto the metasurface via a pipette, and individually left to evaporate under 0–15 kV between the metasurface and ground plate. The distribution of the latices within the evaporative region was then presented (Figure 23 and Figure 24). It was found that the electrophoretic technique enabled a greater number of latices to become trapped by metasurface apertures as the droplets evaporated over that of evaporation at 0 V between the metasurface and ground plate. However, the inability of the Monte Carlo simulations to predict the complex latice accumulations and inconsistencies between adjacent aperture rows show that these models do not fully explain the physics governing the latice accumulation observed in this experiment. Therefore, a physical model which accounts for such differences between experiment and simulation should be further investigated, and a quantitative method of determining the experimental gathering efficiency of the latices be devised.

Then, 5 mL of the 333 ppm solution was deposited over a three inch diameter region of the metasurface and left to evaporate at 0 V (once) and 15 kV (three times) in order to measure the change in resonant frequency and maximum transmittance amplitude of the metasurface due to the presence of the latex nanoparticles. Upon inspection of the large regions after evaporation, very clear differences in the structure and distribution of the latices were observed between the 0 V and 15 kV spots. Namely, it was seen that a much larger fraction of the particles was distributed over the evaporative surface area in the 15 kV cases over that of the 0 V case (Figure 27). Interesting structures were observed within the large evaporative regions at 15 kV which were also observed in the droplet regions, such as tendril-like extensions along metasurface apertures (Figure 25, Figure 26, Figure 27 and Figure 28) which were not present at 0 V. Furthermore, it was seen that the particles preferred to form long strands which connected the metasurface apertures. Upon measuring the transmittance spectra of the 15 kV and 0 V evaporative regions using a W-band system, it was found that the electrophoretic technique allowed shifts in maximum transmittance amplitude and resonant frequency from that of the bare metasurface which were approximately 900–1300%, and 1408–1699% greater than the 0 V case, respectively. It is suggested that the increase in concentration of the latices within the metasurface regions of greatest sensitivity successfully enhanced the sensing capabilities of the metasurface at W-band.

Finally, the possibility of a relationship between the metasurface topology and the gathering efficiency as a function of voltage and particle radius was explored. It was found that one may be able to estimate the maximum theoretical gathering efficiency of a metasurface based upon its geometry and a particular latex particle of interest. It is suggested that the relationship between the gathering efficiency of a metasurface and its topology be further explored, as it may enable one to develop a metasurface which is optimized for use with this electrophoretic technique, for many other types of particles of interest.

In closing, it has been shown that the proposed electrophoretic technique serves as a worthy candidate to improve the sensing capabilities of metasurfaces to low concentration charged particles in solution.

## Figures and Tables

**Figure 1 sensors-23-08359-f001:**
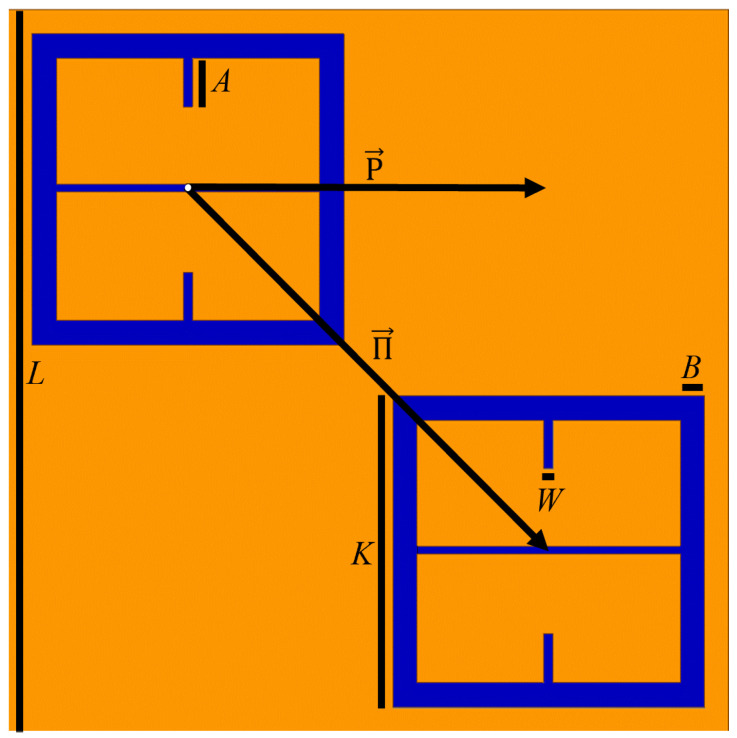
The metasurface unit cell which will be the focus of this paper. Orange is 250 nm thick layer of copper, blue is a 400 μm thick silicon wafer beneath it.

**Figure 2 sensors-23-08359-f002:**
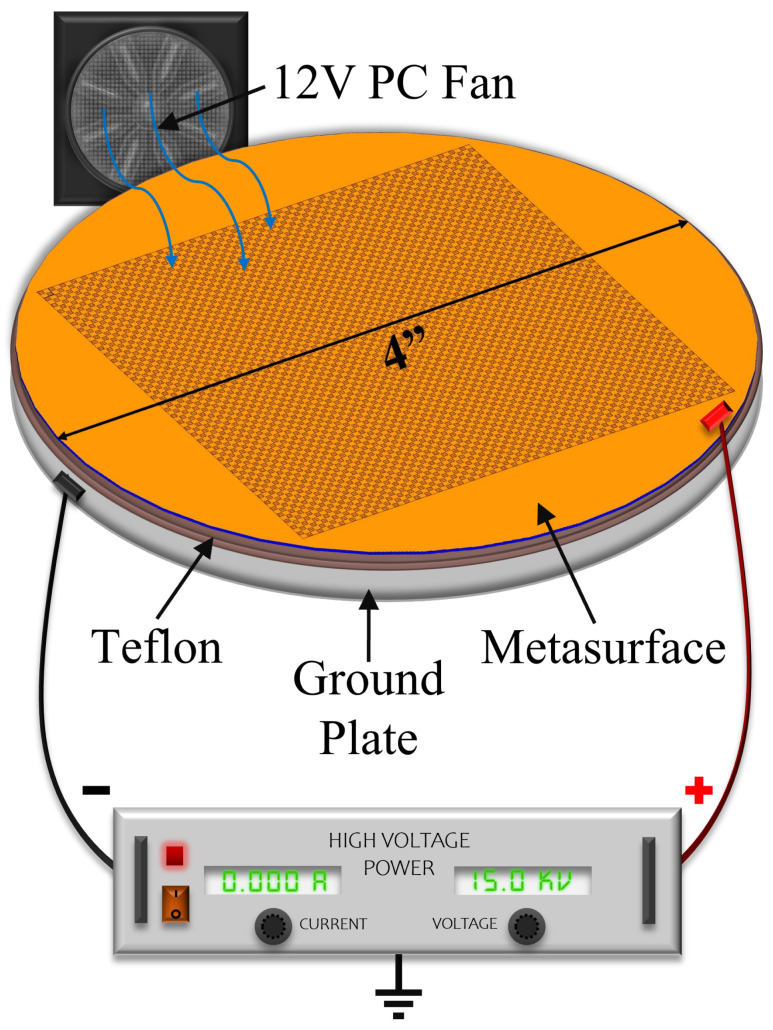
A diagrammatic representation of the experimental apparatus constructed for the purpose of studying electrophopretically-guided particle accumulation at metasurface apertures. The metasurface (copper (orange) etched atop a 400 μm silicon wafer (dark blue)) rests on a grounding plate (grey) which is covered in a thin layer (0.8 mm thick) of Teflon (brown) to prevent electrical breakdown between the metasurface and ground plate. The ground of the high voltage power supply is connected to the grounding plate, and the metasurface is connected to the positive terminal of the high voltage power supply. A small computer fan was placed into the large containment box to aid in evaporation (not facing the solution).

**Figure 3 sensors-23-08359-f003:**
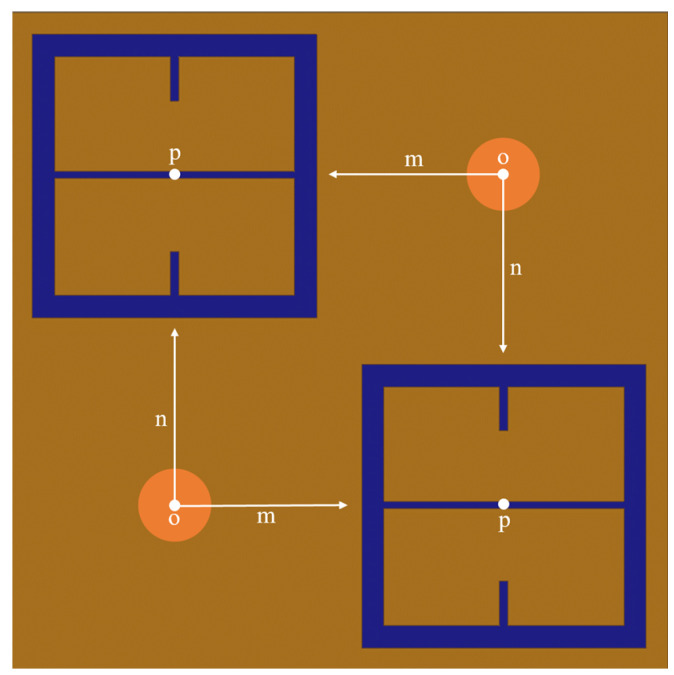
The diagram of the method used to probe the sensitivity region of the metasurface for sensing applications. The light orange circles represent the holes in a FR4 sheet on top of the metasurface. This only depicts a single unit cell from the full metasurface that was simulated.

**Figure 4 sensors-23-08359-f004:**
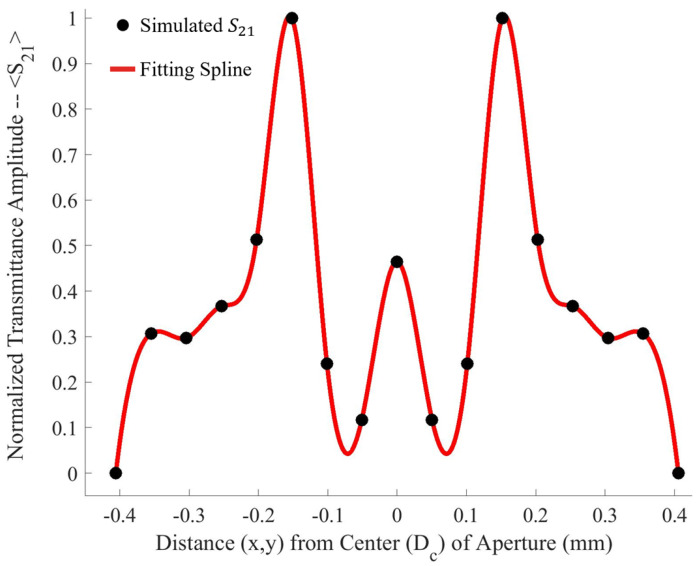
The normalized average maximum transmittance amplitudes recorded after each step in the m and n directions (black dots) with a spline fit between points (red line). In this figure, 0 on the *x*-axis corresponds to point *p* in Figure 3.

**Figure 5 sensors-23-08359-f005:**
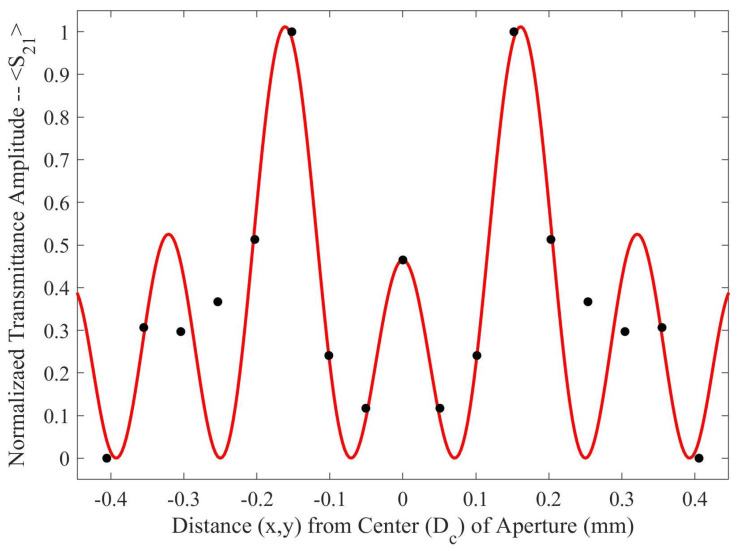
The normalized average maximum transmittance amplitudes recorded after each step in the m and n directions (black dots), compared to the modified Fraunhofer model (red line), presented in Equation (Equation 1). In this figure, 0 on the x-axis corresponds to point p in Figure 3.

**Figure 6 sensors-23-08359-f006:**
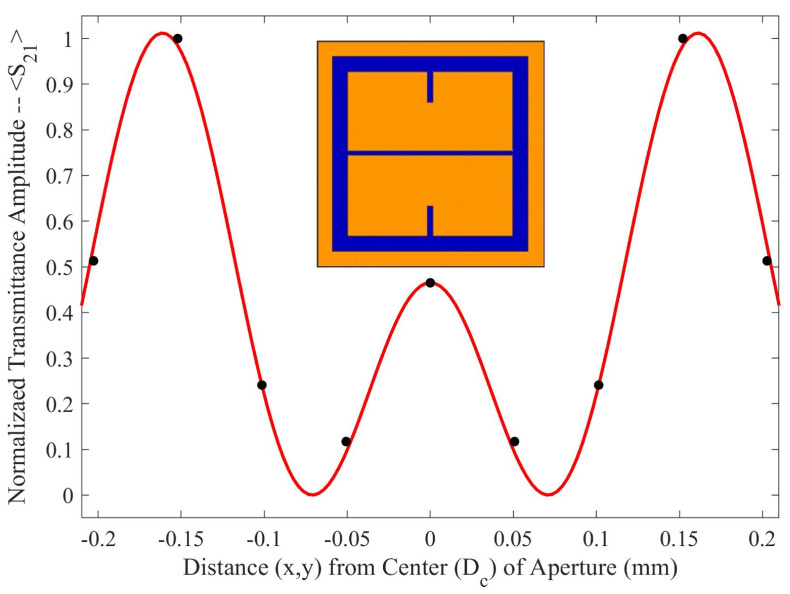
The fit of the modified Fraunhofer model (red line) within the Fraunhofer Accurate sensitivity region (black dots). The metasurface region in which this model is accurate is represented by the meta atom at the center of the figure.

**Figure 7 sensors-23-08359-f007:**
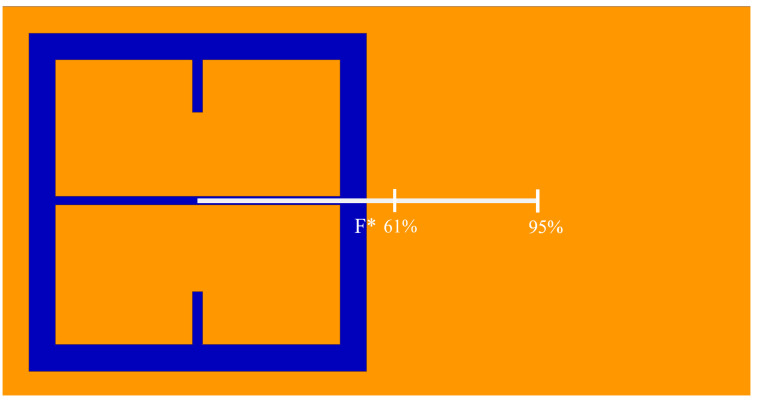
Representation of the distance from the center of the metasurface apertures in which the Fraunhofer Accurate ($F^{*}$) and 95% Sensitivity Regions can be found.

**Figure 8 sensors-23-08359-f008:**
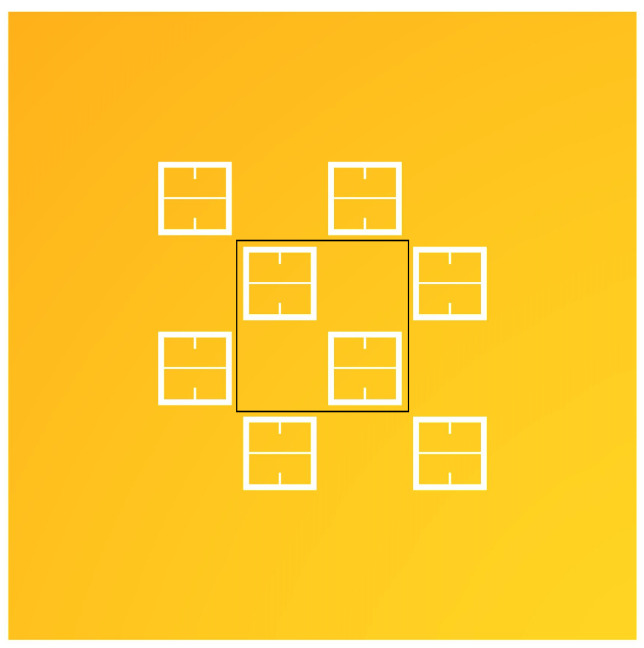
Aerial view of the metasurface region in which subsequent simulations will be conducted.

**Figure 9 sensors-23-08359-f009:**
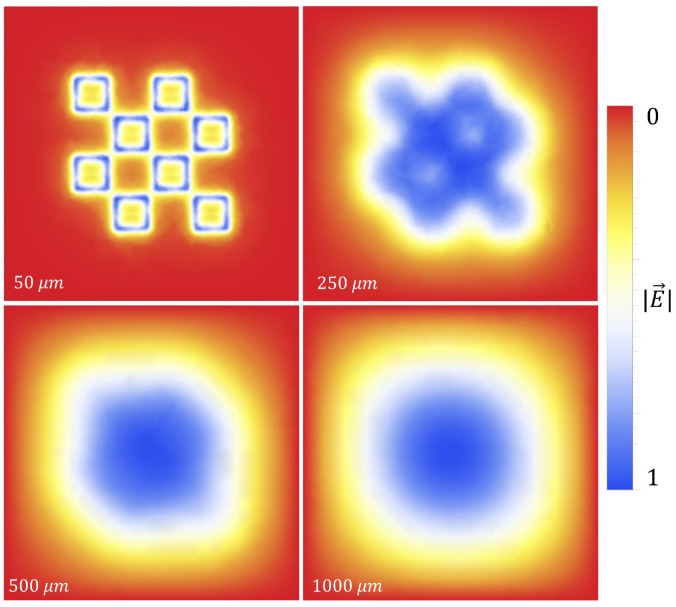
The normalized electric field density produced in a plane parallel to the surface of the metasurface at a height of 50 μm, 250 μm, 500 μm, and 1000 μm.

**Figure 10 sensors-23-08359-f010:**
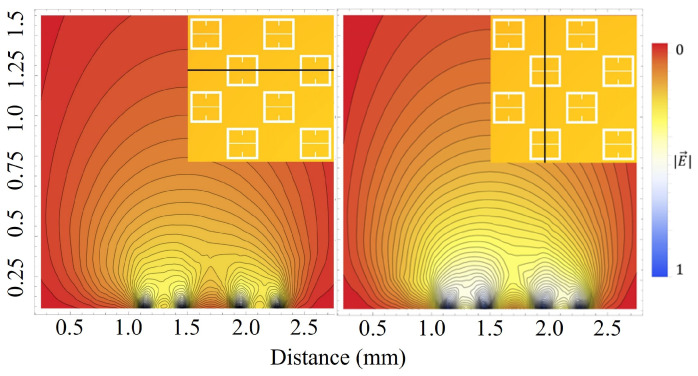
The normalized electric field density and field contours produced in a plane perpendicular to the surface of the metasurface. The black lines within the sub-figures at the top right of the fields represent an aerial view of the plane in which the fields are presented.

**Figure 11 sensors-23-08359-f011:**
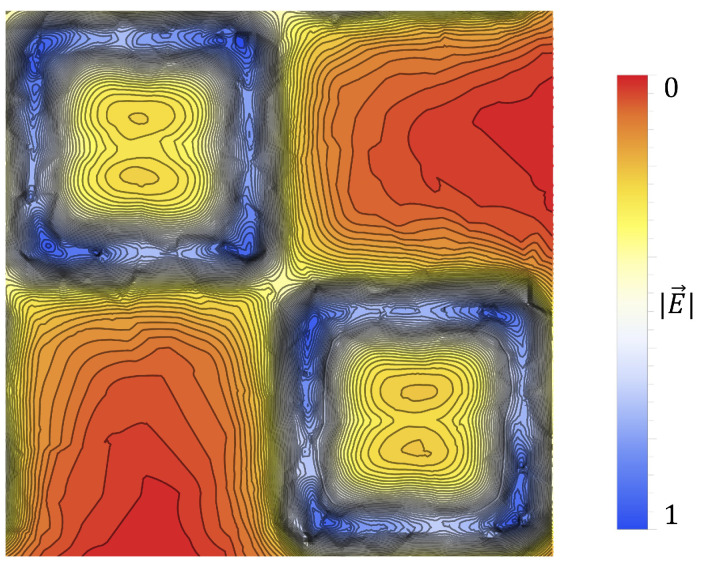
The normalized electric field density and contours produced in a plane parallel to, and 50 μm above the metasurface.

**Figure 12 sensors-23-08359-f012:**
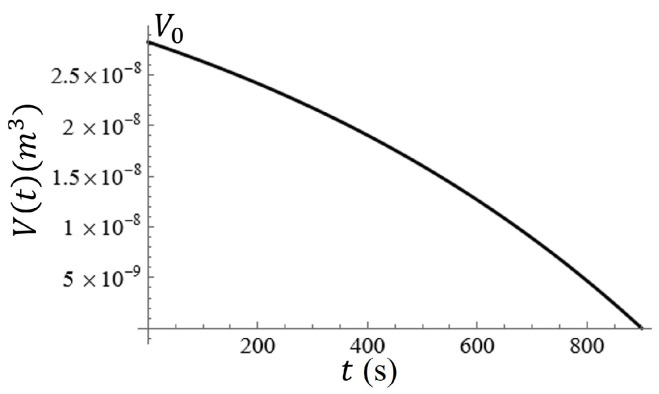
The change in volume as a function of time of a droplet of water in which the colloidal latices were suspended.

**Figure 13 sensors-23-08359-f013:**
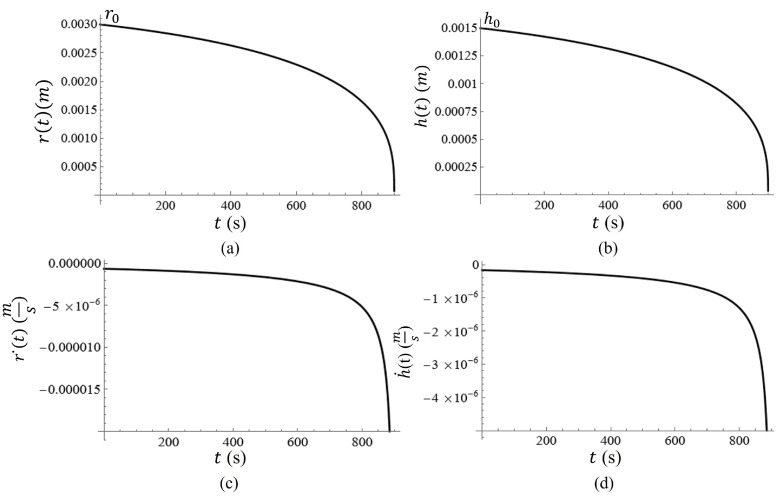
(**a**) The radius as a function of time of a water droplet simulated; (**b**) the height of the water droplet as a function of time; (**c**) the rate at which the droplet radius changes; (**d**) the rate at which the height of the droplet changes as a function of time.

**Figure 14 sensors-23-08359-f014:**
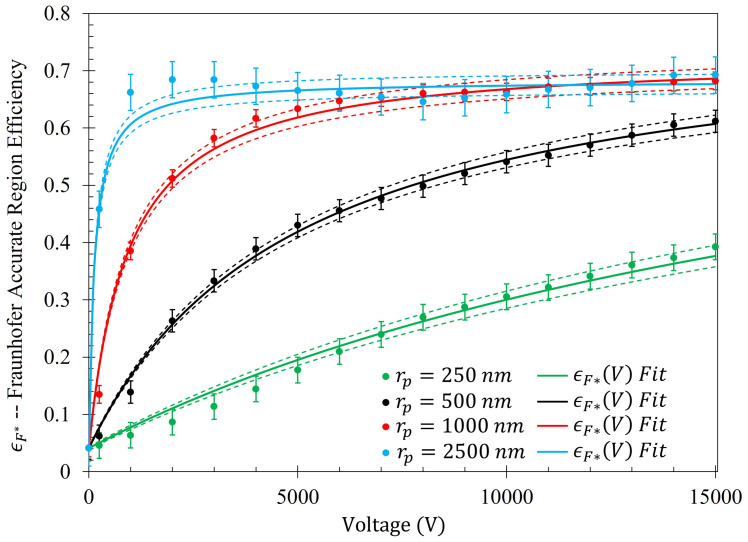
The gathering efficiency of the electrophoretic technique within the Fraunhofer Accurate region (depicted in Figure 7) on the metasurface as a function of voltage. The solid lines represent the fit of the simulated data, each represented by Equation (Equation 14). The dotted lines represent the 95% confidence interval, and the error bars represent the magnitude of the average residuals between the fits and the simulated data (each tail is residual length).

**Figure 15 sensors-23-08359-f015:**
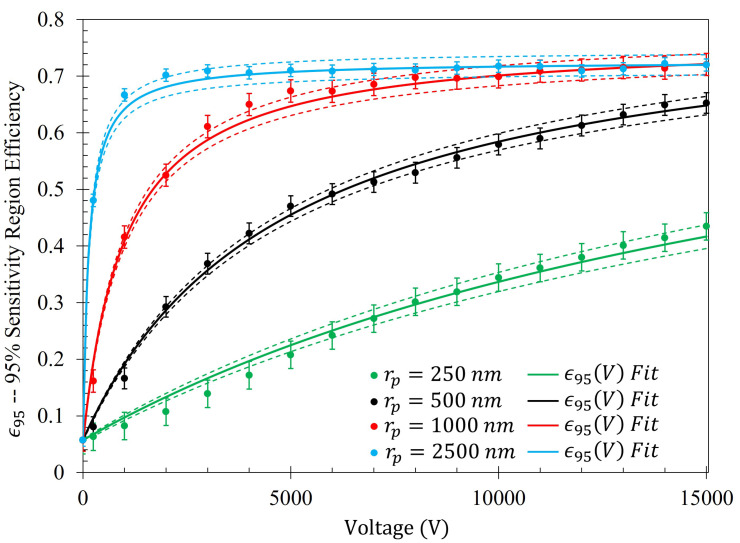
The gathering efficiency of the electrophoretic technique within the 95% Sensitivity region (depicted in Figure 7) on the metasurface as a function of voltage. The solid lines represent the fit of the simulated data, each represented by Equation (Equation 14). The dotted lines represent the 95% confidence interval, and the error bars represent the magnitude of the average residuals between the fits and the simulated data (each tail is residual length).

**Figure 16 sensors-23-08359-f016:**
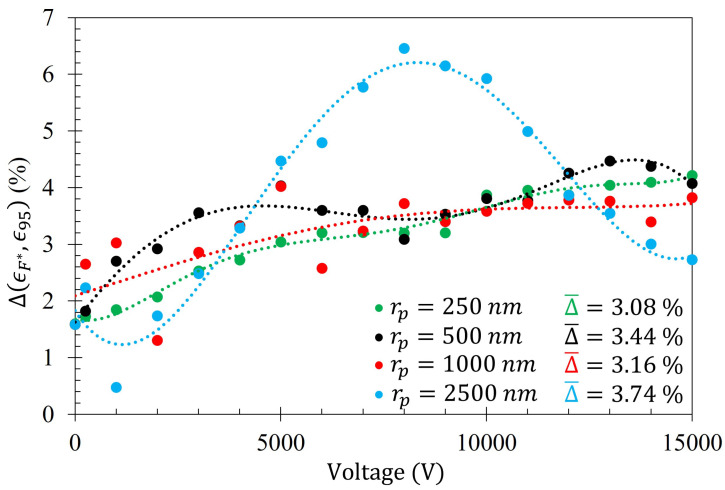
The difference in gathering efficiency of the electrophoretic technique between the Fraunhofer Accurate (61%) and 95% Sensitivity Regions on the metasurface as a function of voltage.

**Figure 17 sensors-23-08359-f017:**
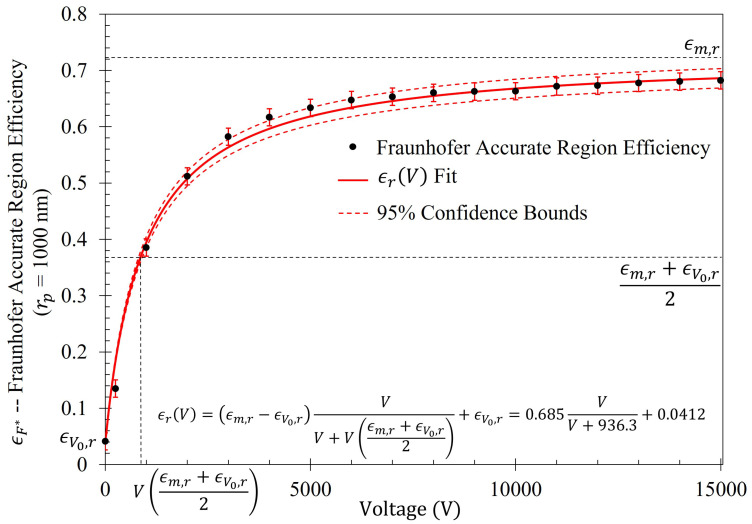
The gathering efficiency of the electrophoretic technique within the Fraunhofer Accurate region on the metasurface as a function of voltage for 1000 μm particles described in Table 3.

**Figure 18 sensors-23-08359-f018:**
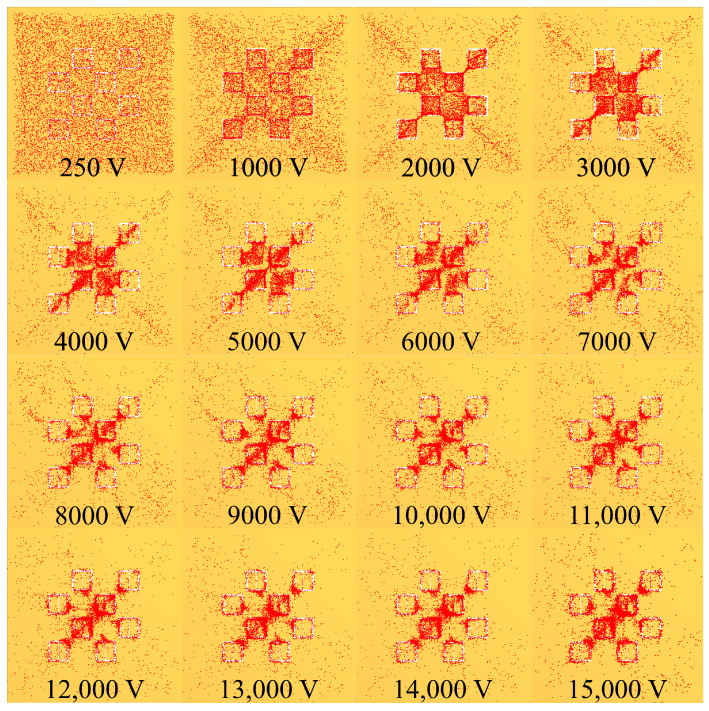
A graphical depiction of the gathering efficiency of the electrophoretic technique using the 500 μm radius particles presented in Table 3, as a function of voltage.

**Figure 19 sensors-23-08359-f019:**
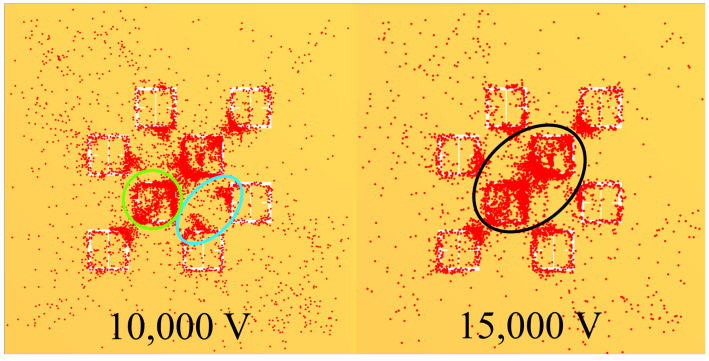
Enlarged features of the 10 kV and 15 kV graphical representations of the gathering efficiency of the 500 μm latex nanoprticles presented in Figure 18.

**Figure 20 sensors-23-08359-f020:**
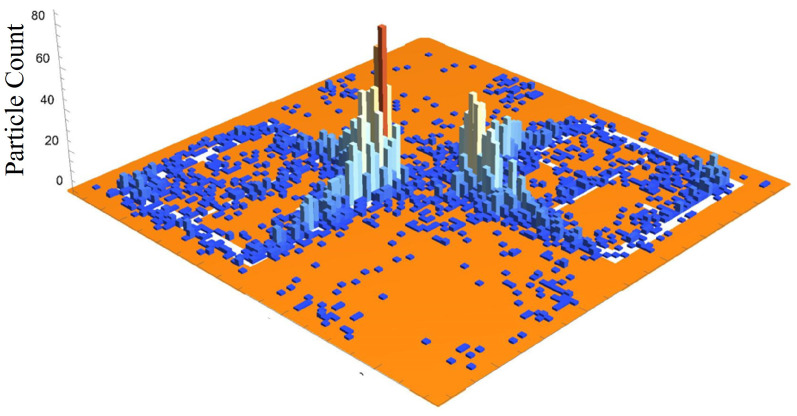
A 3D histogram depicting the packing density of the 500 μm particles within the central unit cell at 15 kV.

**Figure 21 sensors-23-08359-f021:**
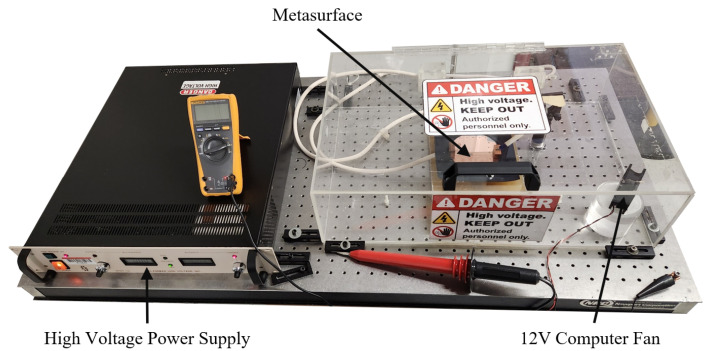
The experimental apparatues designed and constructed for the purpose of testing the electrophoretic technique.

**Figure 22 sensors-23-08359-f022:**
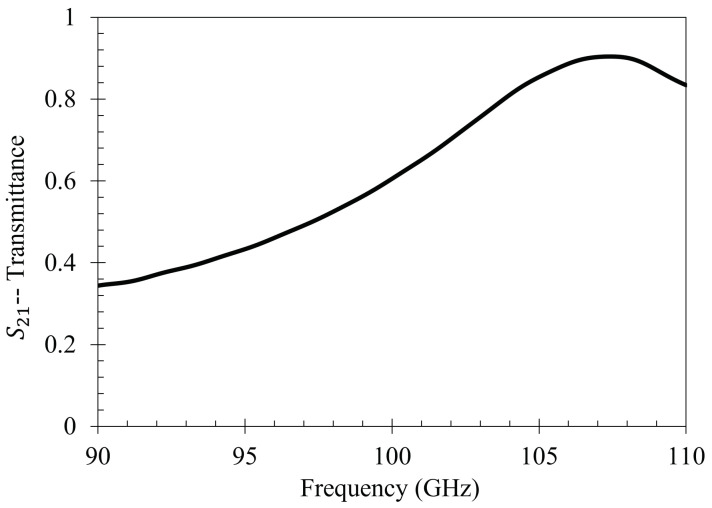
The transmittance spectra of the bare metasurface measured using a Network Analyzer and a pair of W-band horns and lenses.

**Figure 23 sensors-23-08359-f023:**
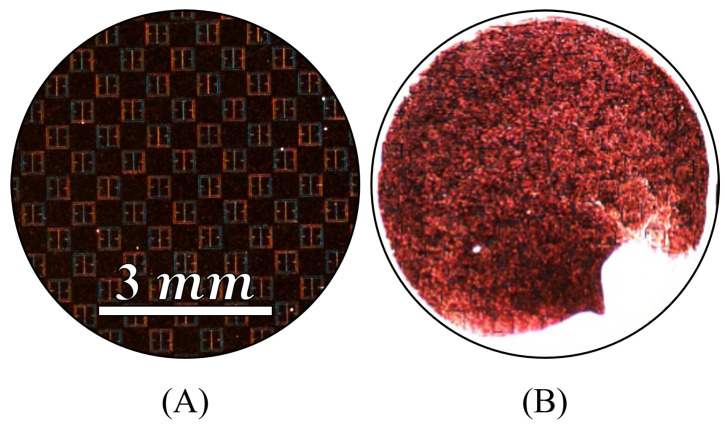
(**A**) inspection scope image of the metasurface using identical exposure settings as seen in (**B**), with no particles present on the surface. (**B**) the inspection scope image of a 333 ppm latex nanoparticle droplet after evaporation with 0 V applied to the metasurface. Latex nanoparticles are any white features within the droplet areas.

**Figure 24 sensors-23-08359-f024:**
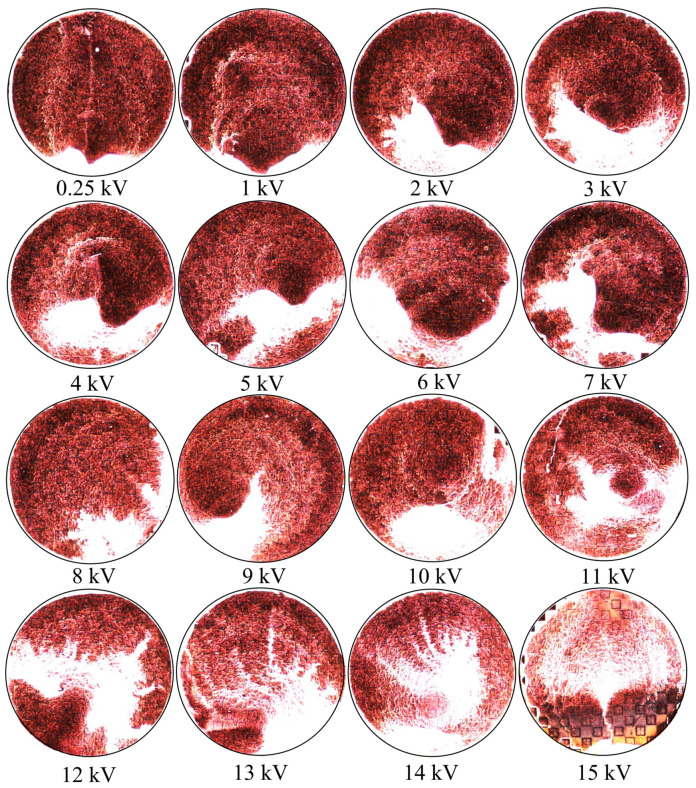
The distribution of particles imaged with an inspection scope within 333 ppm droplet areas after evaporation, while 0 V, 250 V, and 1–15 kV was individually applied to them. The exposure settings and magnification were identical to those presented in Figure 23. Latex nanoparticles are any white features within the droplet areas.

**Figure 25 sensors-23-08359-f025:**
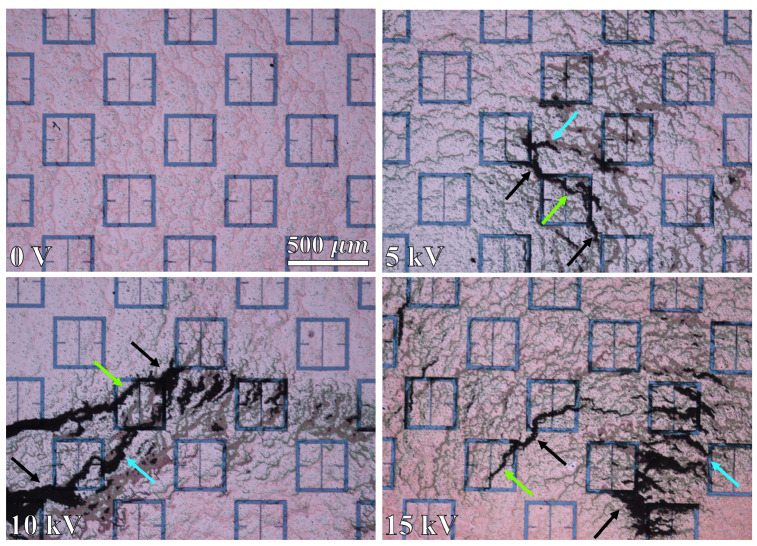
5× microscope images of typical features observed within the 333 ppm droplet regions after evaporation, depicted in Figure 24, at 0 V, 5 kV, 10 kV, and 15 kV. The color of each arrow corresponds to a similarly enclosed oval of the same color in Figure 19.

**Figure 26 sensors-23-08359-f026:**
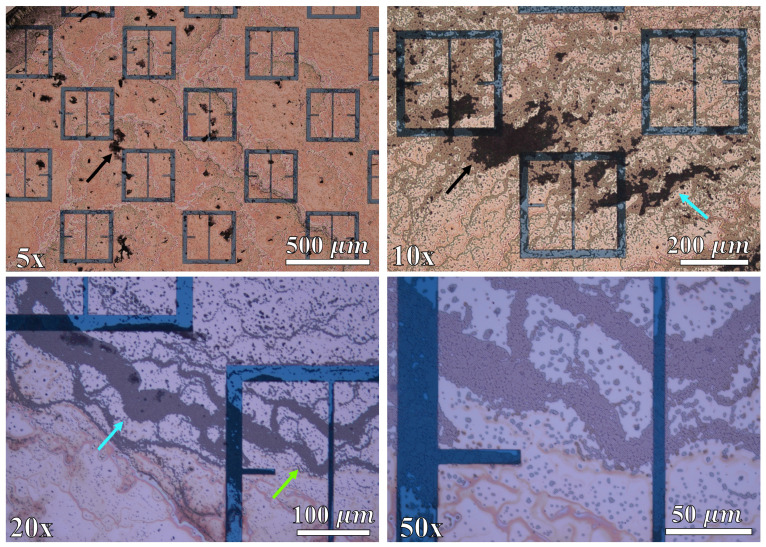
5×, 10×, 20×, and 50× microscope images of regions within a 125 ppm droplet evaporated at 15 kV.

**Figure 27 sensors-23-08359-f027:**
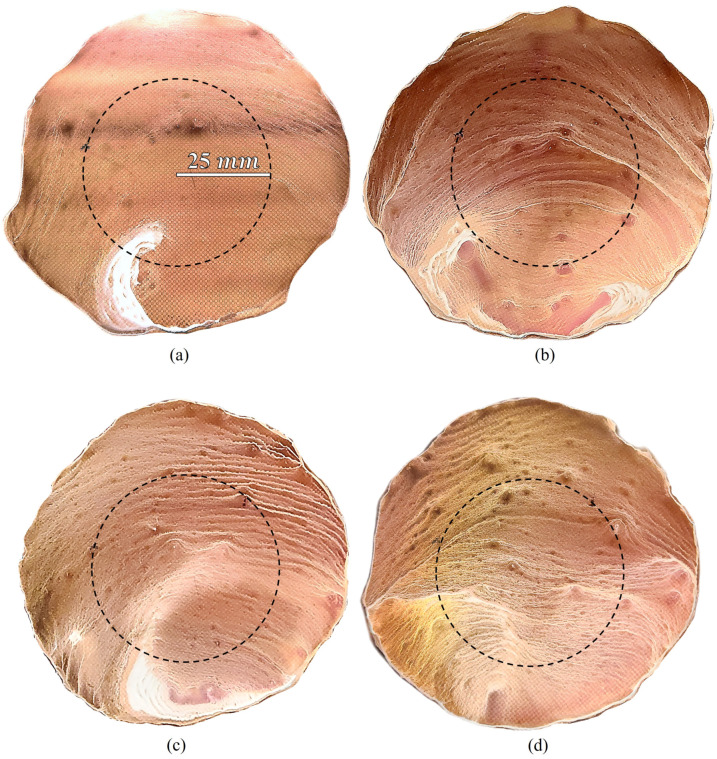
(**a**) the large metasurface region coated in 333 ppm latex nanoparticle solution, and evaporated at 0 V. (**b**–**d**) the large metasurface region covered in 333 ppm latex nanoparticle solution, and evaporated at 15 kV. The black dotted line represents the FWHM of the Gaussian beam within the focal plane, centered on the metasurface/solution area.

**Figure 28 sensors-23-08359-f028:**
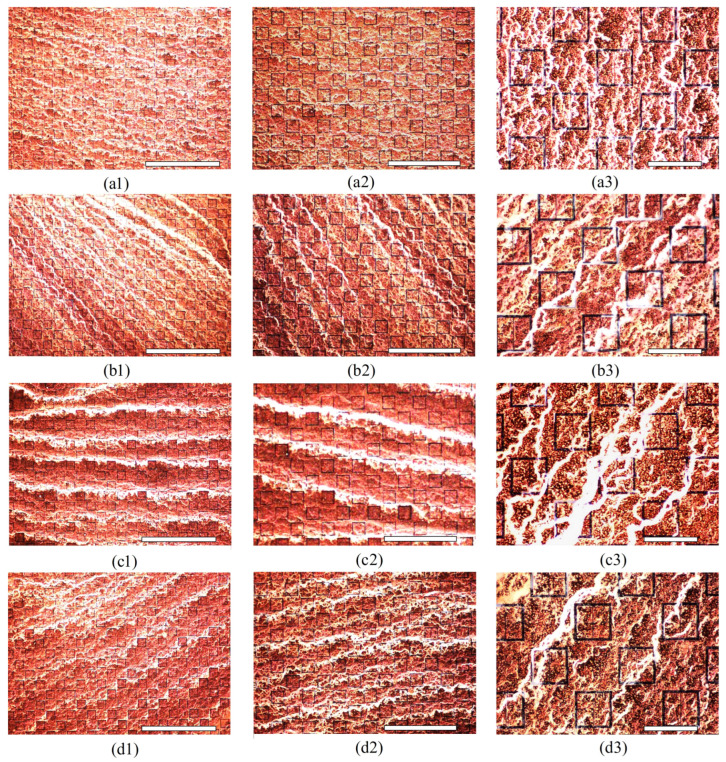
Typical features observed within the large evaporative regions depicted in Figure 27a–d, at different magnifications under an inspection microscope. (**a1**–**a3**) depict some of typical features observed within Figure 27a. (**b1**–**b3**) depict some of typical features observed within Figure 27b. (**c1**–**c3**) depict some of typical features observed within Figure 27c. (**d1**–**d3**) depict some of typical features observed within Figure 27d. Scale bars for (**a1**,**b1**,**c1**,**d1**) are 3 mm. Scale bars for (**a2**,**b2**,**c2**,**d2**) are 1 mm. Scale bars for (**a3**,**b3**,**c3**,**d3**) are 0.5 mm.

**Figure 29 sensors-23-08359-f029:**
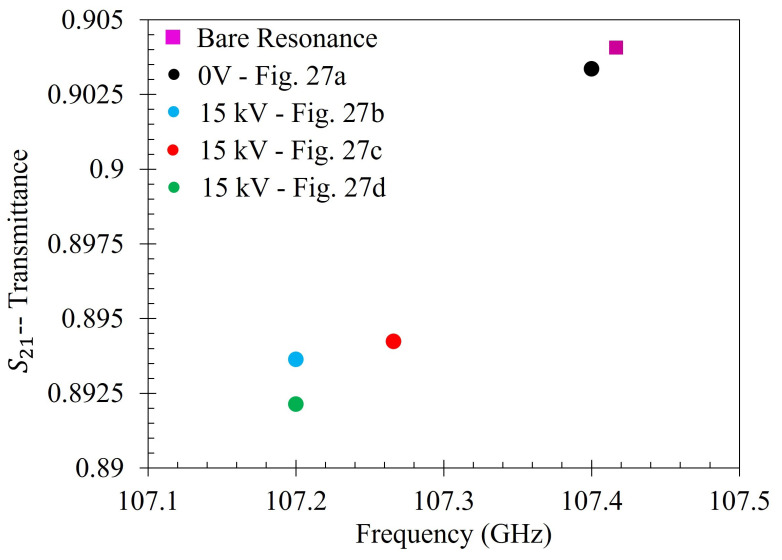
The experimentally measured maximum transmittance amplitude and resonant frequency of the bare metasurface, and after being coated in the 333 ppm nanoparticle solution (as depicted in Figure 27).

**Figure 30 sensors-23-08359-f030:**
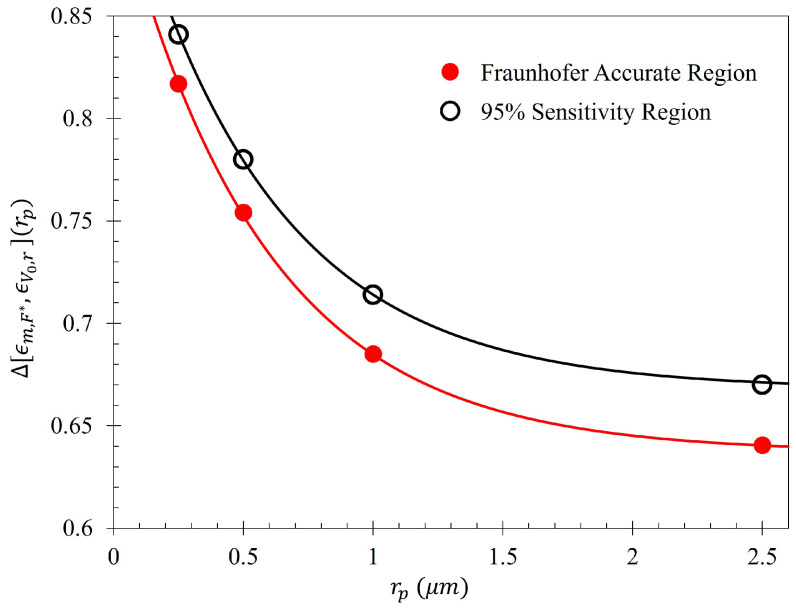
$\Delta \left[\epsilon_{m,r},\epsilon_{V_{0},r}\right]\left(r_{p}\right)$ as a function of particle radius. Equations presented in Table 7.

**Figure 31 sensors-23-08359-f031:**
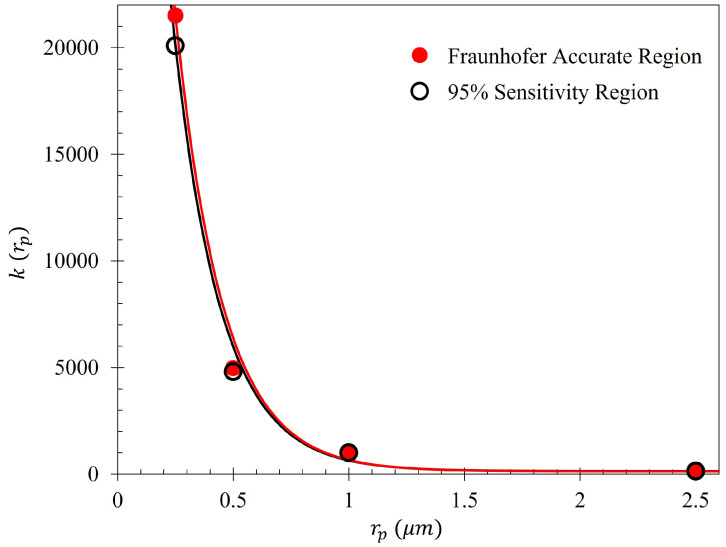
$k(r_{p})$ as a function of particle radius. Equations presented in Table 7.

**Figure 32 sensors-23-08359-f032:**
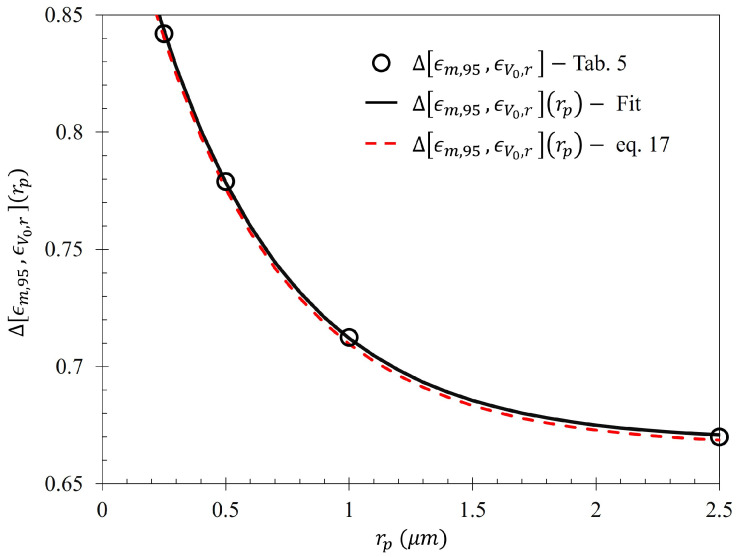
$\Delta \left[\epsilon_{m,95},\epsilon_{V_{0},95}\right]\left(r_{p}\right)$ (from Figure 30) and $\Delta [\epsilon_{m,95},\epsilon_{V_{0},95}]$ (from Table 5) compared to the curve produced by Equation (Equation 17).

**Table 1 sensors-23-08359-t001:** Metasurface design parameters which correspond to those depicted in Figure 1.

Label	Description	Value (μm)
L	Metasurface unit cell length	812
$|\vec{\Pi }|$	Diagonal lattice period	574
$|\vec{P}|$	IBZ ${}^{1}$ wall length	406
*K*	Meta-atom major length	350
*W*	Dipole slot width	9
*A*	Dipole slot length	55
*B*	Meta-atom minor width	28

${}^{1}$ Irreducible Brillouin Zone.

**Table 2 sensors-23-08359-t002:** The lengths of Fraunhofer Accurate and 95% Sensitivity Regions from the aperture centers.

Sensitivity Region	Distance from Aperture Center (μm)
Fraunhofer (61%)	203
95%	358

**Table 3 sensors-23-08359-t003:** The latex nanoparticle parameters used in Monte Carlo simulations of the electrophoretic technique.

Particle Radius (nm)	Mass (kg)	Surface Charge Density ($10^{-14}$ $\frac{C}{\mu m^{2}}$)	Surface Charge (*C*)
250	7.2 · $10^{-17}$	4.2	3.3 · $10^{-14}$
500	5.8 · $10^{-16}$	8.0	2.51 · $10^{-13}$
1000	4.6 · $10^{-15}$	15.5	1.95 · $10^{-12}$
2500	7.2 · $10^{-15}$	38.17	2.99 · $10^{-11}$

**Table 4 sensors-23-08359-t004:** The gathering efficiency of the metasurface at 0 V within the Fraunhofer Accurate and 95% Sensitivity Regions.

Sensitivity Region	Efficiency
Fraunhofer (61%)	0.0412
95%	0.0571

**Table 5 sensors-23-08359-t005:** The equations of best fit which model the gathering efficiency of the metasurface within the Fraunhofer Accurate and 95% Sensitivity Regions presented in Figure 14 and Figure 15.

Particle Radius (nm)	$\epsilon_{F^{*}}\left(V\right)$	$\epsilon_{95\%}\left(V\right)$	$R^{2}$ (Fraunhofer/95%)	$\epsilon_{m,r}$ (Fraunhofer/95%)
250	$0.817\frac{V}{V+21520.9}+0.0412$	$0.841\frac{V}{V+20095.7}+0.0571$	>0.99/0.99	0.859/0.898
500	$0.754\frac{V}{V+4980.9}+0.0412$	$0.780\frac{V}{V+4804.6}+0.0571$	>0.99/0.99	0.795/0.837
1000	$0.685\frac{V}{V+936.3}+0.0412$	$0.714\frac{V}{V+1000.7}+0.0571$	>0.99/0.99	0.726/0.771
2500	$0.641\frac{V}{V+134.3}+0.0412$	$0.670\frac{V}{V+145.0}+0.0571$	>0.99/0.98	0.682/0.726

**Table 6 sensors-23-08359-t006:** The measured relative increase in the shifts in resonant frequency ($\Delta F_{0V,15kV}$) and transmittance amplitude ($\Delta S_{0V,15kV}$) at 15 kV (Figure 27b–d), over that observed at 0 V (Figure 27a). Plot of points presented in Figure 29.

Figure	$\Delta F_{0V,15\mathrm{kV}}$ (%)	$\Delta S_{0V,15\mathrm{kV}}$ (%)
Figure 27b	1300	1486
Figure 27c	904	1408
Figure 27d	1300	1699

**Table 7 sensors-23-08359-t007:** The equations which describe the fits presented in Figure 30 and Figure 31.

Radius Dependent Parameter	Fit	$\lim_{r\to 0}$	$\lim_{r\to \infty }$
$\Delta \left[\epsilon_{m,F^{*}},\epsilon_{V_{0},r}\right]\left(r_{p}\right)$	$0.284e^{-\frac{r_{p}}{0.544}}+0.639$	0.923	0.639
$k_{F*}\left(r_{p}\right)$	$7.33\cdot 10^{4}e^{-\frac{r_{p}}{0.203}}+134$	$7.34\cdot 10^{4}$	134
$\Delta \left[\epsilon_{m,95},\epsilon_{V_{0},95}\right]\left(r_{p}\right)$	$0.277e^{-\frac{r_{p}}{0.544}}+0.668$	0.945	0.668
$k_{95}\left(r_{p}\right)$	$6.84\cdot 10^{4}e^{-\frac{r_{p}}{0.203}}+144.7$	$6.86\cdot 10^{4}$	144.7

## Data Availability

Not applicable.

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
