# Peer review of "A Novel Electrophoretic Technique to Improve Metasurface Sensing of Low Concentration Particles in Solution"

_sensors, 2023, doi:10.3390/s23208359_

Round 1

Reviewer 1 Report

In the manuscript entitled "A novel electrophoretic technique to improve metasurface sensing of low concentration particles in solution" Z.A. Kurland et al. have reported a novel electrophoretic technique to improve the sensing capabilities of charged particles in solution.

First of all, to which type of sensors do the authors refer? Since it is not clear. Please, discuss.

In the abstract, the authors should indicate briefly the motivation of their reported study, and the best obtained results; instead, the abstract is too generic. Moreover, in the introduction, the authors should give more details on the used methodology and on the possible application.

Please, give more examples  of metasurfaces, indicating the type of materials they are made and which nanoparticles are detected; add references.

What does "low concentration" mean? Please, add a concentration range.

Please, define the phases of the electrophoretic process.

Please, define the sensing mechanism.

How do you produce/prepare the metasurface?

Define the latex particles.

How do you estimate the correctness of the sensing proposed method? Discuss. Which is the reference?

Which are the sensitivity and selectivity of proposed method? Discuss.

Compare the proposed method with the others reported in literature.

Add a scheme of the proposed electrophoretic process.

Which is the LOD of the method?

From the inspection images (report the techinque used to acquire the images), it seems to be a deterioration of the metasurfaces during the process. Discuss.

Report a chemical-physic mechanism to explain the interaction between the metasurface and the particles.

From the reported inspection images, it seems to be a problem of reproducibility and stability of measurements. Please, discuss.

In my opinion, the manuscript can be accepted with major revisions.

Reviewer 2 Report

The authors propose an electrophoresis technique to improve the sensing ability of charged particles in solution, and simulate and experimentally test the proposed technique using latex nanoparticles in solution. The study is comprehensive and the data are detailed and complete.

However, the reviewer found that the literature cited was relatively old and the latest research reports were fewer. This cannot help but make people doubt the novelty of the work described in this paper, and it is recommended that the author refer to recent research reports in the revised draft for comparison, which fully demonstrates the innovation of the work in this paper.

The manuscript mentions that with the proposed technique, at 15 kV, the resonant frequency of the bare antenna is approximately 900%-1300% higher than the offset observed at 0 V, and the maximum transmittance amplitude is 1408%-1699% higher. It is recommended that the revised draft add the innovative points or advantages of this research compared to the relevant work of others, rather than the previous work of one's own team.

In addition, Reference 11 is not used in the text. Please add the quote or remove it. Reference 25 appears earlier in the text than References 21-24. References are numbered in the order in which they appear in the text. Please check the full text and revise it.

Minor editing of English language required.

Round 2

Reviewer 1 Report

The revised manuscript is now in better shape to be accepted for publication.

The English language in this manuscript is of poor quality. The authors must revise their manuscript to remove grammatical errors.